

# The climate of the Eastern Mediterranean and the Nile River basin 2000 years ago using the fully forced COSMO-CLM simulation

Mingyue Zhang[1], Eva Hartmann[1], Sebastian Wagner[2], Muralidhar Adakudlu[3], Niklas Luther[3], Christos Zerefos[4], Elena Xoplaki[1, 3]

[1]Climatology, Climate Dynamics and Climate Change, Department of Geography, Justus Liebig University of Giessen, Giessen, 35390, Germany
[2]Helmholtz-Zentrum Hereon, Geesthacht, 21502, Germany
[3]Center for International Development and Environmental Research, Justus Liebig University of Giessen, Giessen, 35390, Germany
[4]Research Centre for Atmospheric Physics and Climatology, Academy of Athens, Athens, Greece

*Correspondence to*: Mingyue Zhang (Mingyue.zhang@geogr.uni-giessen.de)

## Abstract

Understanding the past climate at regional scale, the impact of natural variability and sensitivity by studying the underlying dynamics and processes, can provide a point of reference for future climate conditions under anthropogenic forcing. The Eastern Mediterranean (EM) and Nile River basin (NR) regions are of particular interest for the study of past climate due to their location under the influence of major atmospheric teleconnections. We developed a high-resolution regional model for paleoclimate applications, COSMO-CLM, by integrating all external forcings and conducted a transient simulation from 500 BCE to 1850 CE. Principal Component Analysis (PCA) was applied for winter/summer precipitation and temperature to validate the model set up and showed very good agreement between simulated and observational/reanalysis data. Further, 400-362 BCE and 1800-1850 CE have been selected for the comparison of the mean climate conditions of the early Roman period (ERP) and pre-industrial times (PI). The comparison of temperature and precipitation suggests comparable mean climatic conditions with spatial differences in terms of variability within the study regions. Over the Eastern Mediterranean (EM), ERP is wetter and warmer in both winter and summer compared to PI, with higher variability in temperature and precipitation in summer than in winter. In the Nile River basin (NR), ERP summers were wetter and more variable compared to PI. The ERP over NR is warmer by approximately 0.5 °C in winter and cooler by 0.5 °C in summer, with low variability in winter and high variability in summer compared to PI. The relevant large-scale circulation of the two periods shows consistent spatial structures with the corresponding precipitation/temperature EOF patterns, albeit with varying amplitudes. The 2500 years transient simulation sheds light to the paleo climate conditions and relevant atmospheric circulation as well as processes of periods of interest in complex areas with detailed output and comprehensive forcing allowing for better representation of the regional climate variability and change. Comparison of simulated output with proxy records, reconstructions and detailed studies of specific events, e.g., volcanic eruptions, can help to capture the spatiotemporal extent of these events and their impact on climate variability and change, in addition to providing insights into their impact on societal change and human history.



## 1 Introduction

The global climate undergoes significant changes, characterized by rising temperatures, more intense, more frequent and more persistent extreme weather and climate events and shifting precipitation patterns (Hennessy et al., 2022). Anthropogenic climate change poses a great threat to ecosystems, economies, and human well-being worldwide (Pörtner et al., 2022). The historical context is critical when assessing present-day climate anomalies, attributing them to forcings and making statements regarding their frequency and severity in a long-term perspective (e.g. Luterbacher et al., 2016). Furthermore, the availability of paleoclimatic data spanning centuries to millennia is a vital resource for studying and characterizing climate changes, offering insights that enhance our comprehension of climate variability, trends, extremes, and contributing essential information for climate mitigation and adaptation strategies (Luterbacher et al., 2016; Luterbacher and Pfister, 2015; Haldon et al., 2014, 2018).

Studying climate at regional and local scales improves our understanding of the dynamical and physical processes involved. In order to prepare useful climate projections and adequate responses to the impacts of climate change on vulnerable areas (Kelley et al., 2015), a better understanding of the past climate at a high spatial resolution is of great importance. The Eastern Mediterranean (EM) and the Nile River Basin (NR) are located in an area of great climatic and societal interest with a very long history, with important civilizations inhabiting the area for several millennia. Today, the EM area faces substantial societal challenges, partially connected with climate change impacts, including migration and societal disruptions (Lange et al., 2020). The region also represents a prominent climate change hotspot with exceptionally intense warming that exceeds the continental and global averages, while the area experiences an increasing amount of extreme weather events such as heatwaves, droughts, dust storms, heavy precipitation and floods (Zittis et al., 2022).

EM is influenced by mid-latitude, subtropical and tropical weather systems (Alpert et al., 2005), which can lead to a range of extremes, including windstorms, hydrological and temperature extremes (Hochman et al., 2022). Situated in a transition zone between subtropical and mid-latitude climates and located at an atmospheric crossroad, the area is directly influenced by a variety of atmospheric circulation patterns and meteorological processes on different continents (Zittis et al., 2022). For example, in the summer the South Asian Monsoon has a major impact on the Eastern Mediterranean, while in winter, the area is affected by the variability of continental circulation anomalies linked to the Siberian High-Pressure System (Cramer et al., 2018; Paz et al., 2003). The extreme weather over EM, such as heavy precipitation, is mainly governed by the large-scale atmospheric circulation and its interaction with regional synoptic systems (i.e., Cyprus Lows, Red Sea Troughs, Persian Troughs, "Sharav" Lows) and high-pressure systems. Complex orographic features further play an important role in the generation of extreme weather (Hochman et al., 2022). Multiple atmospheric circulation patterns affect the NR region and the water availability of the huge drainage basin that crosses different hydroclimatic zones. The precipitation regime along the Nile catchment is mainly related to the West African monsoon, which itself is modulated by the Indian monsoon dynamics (Ménot et al., 2020). Over the central Ethiopian Highlands, the source region of the Blue Nile, 70% of the annual precipitation falls during summer (June-September) when the region is affected by the rain belt of the Intertropical Convergence Zone (ITCZ) and tropical convection clusters over the continental areas (Conway, 2000). The climate of the region is also influenced by monsoon systems that prevail around Lake Victoria and sections of the Ethiopian highlands (Camberlin, 2009). The moisture sources that actually affect the NR Basin originate from the Gulf of Guinea, the Indian Ocean and the northern inflow from the Mediterranean Sea and the Red Sea (Viste and Sorteberg, 2013). Most of the wet months in Ethiopia occur in connection with enhanced moisture transport from the north (Viste and Sorteberg, 2013).

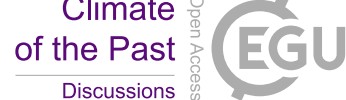

As in the past, the Nile River is also today the main agricultural and economic water resource for multiple African
countries (Singh et al., 2023). Among those, Egypt has always been heavily dependent on the Nile River flow, as one of
the Ancient World "*hydraulic civilizations*" (Singh et al., 2023) and thus provides a unique laboratory to study societal
vulnerability and response to climate variability (Manning et al., 2017). For example, the Nile floods had significant
impacts on the Egyptians' prosperity, thus various strategies have been made to mitigate, adapt to and take advantage of
the impacts of flooding, including technological advancements in water-lifting machines and innovative agricultural
practices.
The study and better understanding of the climate of the past, its variability, the occurrence of extremes and their
interaction with societies are of great significance and scientific interest. It is crucial for gaining insight into current and
potential future environmental challenges, and understanding societal and cultural changes (Xoplaki et al., 2016, 2018).
The Eastern Mediterranean and northeastern African areas offer a relatively dense network of natural archives and
documentary evidence covering the past 2,000 years and indicate the heterogeneous and variable in time and across space
climate of the past two millennia (Xoplaki et al., 2016, 2018, 2021; Zittis et al., 2022).
The Paleoclimate Modelling Intercomparison Project (PMIP) aims at understanding how the climate system responds to
various climate forcings for the documented climate conditions that might differ between historical and current times
(Kageyama et al., 2018). At its fourth phase, PMIP4 focuses as well on the comparison between climate reconstructions
(based on physical, chemical or biological records) and climate modelling results addressing the ability of state-of-the-art
numerical models to realistically simulate climate variability and change of the past, and whether their response to
different external forcings is compatible with paleoclimatic evidence (Kageyama et al., 2018). Paleoclimate studies in the
Eastern Mediterranean and the Nile Basin mostly rely on proxy records (García-Herrera et al., 2007), climate
reconstructions (Luterbacher et al., 2016), and global paleoclimate models with a coarse spatial resolution. The purpose
of regional climate models (RCMs) is to refine climate data from coarse-resolution global climate models (GCMs). By
doing so, RCMs offer more detailed information at smaller, sub-GCM-grid scales. This increased resolution is particularly
valuable for studying regional phenomena and for conducting vulnerability, impacts, and adaptation assessments (Giorgi,
2019). Various studies in the field of present-day regional climate modelling (Worku et al., 2018; Alemseged and Tom,
2015; Bucchignani et al., 2016) demonstrated that the RCMs outperform their global counterparts in various aspects,
mostly related to small-scale meteorological phenomena and the hydrological cycle. Armstrong et al. (2019) found that a
regionally limited version of a high spatial resolution atmospheric GCM can simulate more accurately pre-industrial
climate and enhance the representation of anomalies in certain past atmospheric processes compared to lower resolution
GCMs. By comparing GCM and RCM simulations for different periods in the past, Gomez-Navarro et al. (2012)
concluded that their regional model improves the skill to reproduce high-frequency climate patterns over parts of the
Mediterranean for the last millennium by better mimicking regional circulation patterns and local climate and hence
reducing the biases of the driving GCM. Studies have also highlighted the more realistic representation of topography
and the regional climate information that is valuable for paleoclimate studies (Renssen et al., 2001). Considering that the
existing proxy and observational data over the study region are geographically restricted over specific parts of the area
and the existing GCMs are too coarse, RCMs can help to close this gap by improving the spatial resolution (Bray and von
Storch, 2016).
Climate variations are also related to changes in external forcing parameters (e.g., orbital, solar, volcanic, land use and
greenhouse gases (GHG) (Gómez-Navarro and Zorita, 2013). The climate system is primarily driven by solar radiation,
and the variations of solar irradiance can lead to changes at decadal to centennial time scales (Gray et al., 2010). Changes



in orbital parameters such as eccentricity, obliquity, and precession also have important effects on the latitudinal and
seasonal distribution of solar radiation at different temporal scales (Ludwig et al., 2016; Cubasch et al., 2006). As far as
the volcanic forcing is concerned, of the impact of volcanic sulfate aerosols that are injected into the atmosphere by large
tropical volcanic eruptions, leads to pronounced stratospheric warming and surface cooling (Robock, 2000; Crowley et
al., 2008). Historical studies have matched the Nile flooding with the impact of volcanic eruptions (Manning et al., 2017).
Greenhouse gases (GHGs) trap in turn the longwave radiation that is emitted by the Earth surface and lead to a continuous
increase of energy in the climate system and effects on the global climate (Ramanathan and Feng, 2009). Furthermore,
land use and land cover changes can affect climate by directly altering the surface solar and longwave radiation and
indirectly impacting atmospheric turbulence (Pielke Sr et al., 2011; Zhang et al., 2021). When studying palaeoclimate in
EM and NR, the climate models used are mainly GCMs, and these forcings are introduced in the GCM simulations. For
example, the MPI-ESM-LR which we are using as the input data for the RCM, is fully forced with solar, orbital, volcanic,
GHG and land use change (Jungclaus et al., 2017), but those forcings are not yet fully implemented in the RCM, especially
for the paleoclimate simulation. Studies have shown that implementing some of the forcings into RCM can have a more
accurate representation of regional climate variability (Ludwig et al., 2016, 2017). Hence, the respective changes in
external forcings such as solar, orbital, volcanic, GHG and land use change must be implemented into the COSMO-CLM
to make the RCM more realistic and consistent with its driving GCM.
Thus, we have, to our knowledge, developed the first highly resolved, fully forced, transient paleo regional climate
simulation with COSMO-CLM for the period from 500 BCE to 1850 CE. The implementation of the forcings and
sensitivity experiments are described in Hartmann et al. (submitted). A temporally and spatially highly resolved, regional
simulation across the complete study area (Eastern Mediterranean and Nile River Basin, EMNR) allows the study of
regional-to-local paleoclimate processes with higher accuracy. Further, it enables the exploration of the association
between the regional climate patterns and the large-scale atmospheric circulation patterns from the GCM world. An asset
of the simulation is its contribution to the study of the impact of the occurrence of extreme climate conditions on societies
through interdisciplinary, collaborative research.
We present and exploit the advantages and breakthroughs of the fully forced, transient simulation by assessing the ability
of the simulation to represent the climatic conditions of the area in space and time and the links with the large-scale
circulation. The assessment is divided into two approaches, an evaluation during the present time and a comparison of
two periods, one in the first millennium BCE and one in the pre-industrial times.
The paper is structured as follows: The section "Data and Methods" describes the RCM COSMO-CLM, the reanalysis
and observation time series and provides information on the implemented methods. In the section "Results and
Discussion", the findings of the analysis are presented and interpreted in detail. Finally, the "Conclusions" section
summarizes the main findings and their significance and briefly discusses possible directions for future research.
**2 Data and methods**
**2.1 Regional Climate Model Simulations**
The COSMO-CLM (COnsortium for Small-scale MOdelling in CLimate Mode - CCLM) (Rockel et al., 2008) is a widely
used RCM that has been used to investigate climate change under different forcings, such as land use, orbital and GHG.
The CCLM is originally designed to perform simulations from 1850 CE onward and its performance largely depends on
the driving GCM (Armstrong et al., 2019). Multiple CCLM simulations have been recently performed at different



resolutions up to very high convection-permitting resolution (Raffa et al., 2023) in the frame of the CORDEX
(Coordinated Regional Climate Downscaling EXperiment, https://cordex.org/) initiative.
In this work, we revolutionize by implementing the CCLM in model version 5.0 with CLM version 16 to an adjusted
paleoclimate version, fully forced with orbital, solar, GHG, volcanic and land-use changes (Hartmann et al., submitted).
We performed a transient simulation from 500 BCE to 1850 CE at the Deutsches Klimarechenzentrum (DKRZ), which
is forced with the new MPI-ESM-LR simulation "Mythos" that is performed under the CMIP6 protocol after 1 AD (MPI-
ESM-LR 'past2k') (Jungclaus et al., 2017) at ~1.875° resolution. The period 500 BCE until 1 BC is according to the
external forcings used by Bader et al., (2020) in a simulation for the entire Holocene.
The implemented forcings in COSMO-CCLM are identical to those of MPI-ESM-P "Mythos" simulation, namely, the
orbital forcing is represented by the eccentricity, the obliquity and the longitude of perihelion (Berger, 1978) and the solar
forcing by the total solar irradiance (Jungclaus et al., 2017). The changes in GHG concentrations consist of $CO_2$, $CH_4$ and
$N_2O$ (Meinshausen et al., 2017), while the volcanic forcing is based on the stratospheric aerosol optical depth (AOD) at
550 nm wavelength by Toohey and Sigl, (2017).
The interpolation of the driving data to the model is done with INT2LM in version 2.05 with CLM version 1
(INT2LM−v2.05clm1) (Schättler and Blahak, 2017). The time integration is the two-step Runge-Kutta scheme (Jameson
et al., 1981) with a 300 seconds time step. The convection parameterization based on the Tiedtke scheme (Tiedtke, 1988)
is used. The representation of albedo and aerosols are found to be the most important parameters in the studied region
(see also Hartmann et al., submitted) and are set the same as in Bucchignani et al., (2016). The land surface model is
TERRA-ML (Doms et al., 2011; Schulz et al., 2016). The simulations are carried out in a domain including the Eastern
Mediterranean, the Middle East and the Nile River Basin from Lake Victoria to the Nile Delta (Fig. 1, 4° − 60° E, 5° S −
49° N). For the analysis of the precipitation, we consider the areas of the Eastern Mediterranean (EM, Fig 1b) and the
Nile River Basin (NR, Fig. 1c) separately as the precipitation regimes are significantly different between the two regions.
While for the analysis of temperature, we have selected a region, include the Eastern Mediterranean and Nile River basin
(EMNR, Fig. 1a).
The following periods have been used for the assessment of the fully forced, transient CCLM simulation: i) present period
(1980-2018) to validate the adjusted COSMO-CLM, ii) the pre-industrial period (PI; 1800-1850 CE), and iii) the Early
Roman Period (ERP; 400-362 BCE) for the comparison of the mean climate conditions of the two periods and changes
in the climate characteristics of the study region.

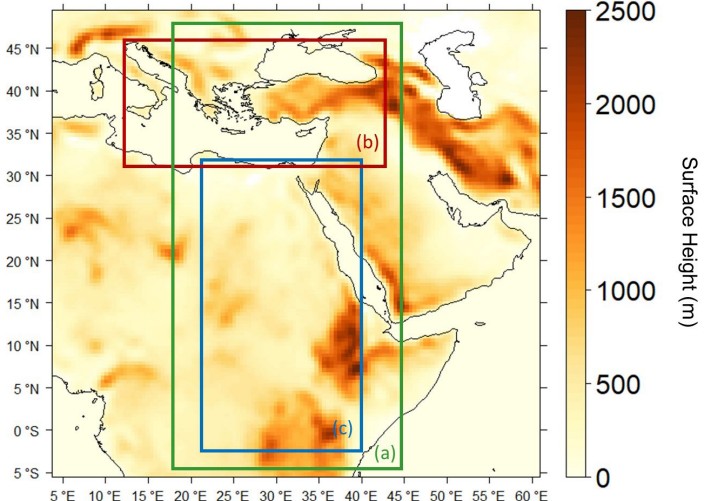

**Figure 1: Topography of the study regions (a) EMNR: indicated with green rectangle, (b) EM: indicated with red rectangle, (c)**
**NR: indicated with blue rectangle. Area (a) is used for the analysis of temperature and the other two areas (b, c) are used for**
**the analysis of precipitation.**
**2.2 Observational Time Series and Reanalysis Data**
We validate the CCLM output, with different observational and reanalysis data sets at 0.5° resolution for the present
period, 1980-2018: the monthly mean temperature and total precipitation from CRU TS v.4.05 (Harris et al., 2020), the
precipitation climatology from the Global Precipitation Climatology Centre V.2020 (GPCC) (Schneider et al., 2022); and
finally the ERA-Interim temperature (Dee et al., 2011) as an additional comparison data set. We restrain from using the
ERA-Interim precipitation due to its dependence on GPCC observations. The CRU TS is the reference data set of the
validation of the present CCLM simulation output.
**2.3 Methods**
**2.3.1 Principal Component Analysis**
Principal Component Analysis (PCA), also referred to as empirical orthogonal functions (EOFs), is a frequently applied
technique in climate sciences and multivariate datasets to reduce the dimensions of the dataset and find a smaller number
of independent variables conveying as much of the original information (w.r.t. to the variance) as possible (Wilks, 2011).
The technique enables the investigation of spatial variability modes by detecting structures in the data (Gallacher et al.,
2017; Bader et al., 2020).
Eq. (1) shows the covariance matrix $C_{XX}$ (where $XX$ represents the detrended seasonal anomalies of temperature and
precipitation) is decomposed according to
$$C_{XX} \cdot e = \lambda \cdot e \qquad (1)$$

This decomposition includes a restructuring of the covariance matrix $C_{XX}$ with the most important patterns (eigenvectors,
$e$) and their principal components (PCs) showing the highest amount of spatial variance (represented by the eigenvalue $\lambda$)
within the full fields. The classic patterns, EOFs, are orthogonal, hence, the eigenvectors are uncorrelated. For some
applications, this is a useful characteristic (i.e., setting up multiple regression models with predictors that are not



collinear). In meteorological applications, however, the orthogonality constraint may be disadvantageous, because most
processes in the real world are not orthogonal (Storch and Zwiers, 1984). We thus applied the VARIMAX rotation to
obtain rotated EOFs ($re_j$, REOFs) that are physically more consistent than the non-rotated patterns. REOFs are thus used
for the validation of the model set-up in the present period (1980 - 2018 CE), while non-rotated EOFs are implemented
for the comparison of the mean climate conditions in PI and ERP times.
The PCs provide the temporal evolution of the (rotated) eigenvectors $re_j$ and are calculated by projecting the original
seasonal anomaly precipitation and temperature time series $X$ onto the (rotated) eigenvectors $re_j$ (REOFs), see Eq. (2):
$$PC_j = \ <X \,|\, re_j>\ \tag{2}$$

where $<\,|\,>$ denotes the dot product and $j$ the index of the according principal component and rotated eigenvector,
respectively.

**2.3.2 Validation of the Model Set-up (1980-2018)**

For the comparison between observed/reanalysis and simulated temperature and precipitation in the present time (1980-
2018), we want to compare spatially homogeneous climatic regions in the study area of the simulated and observed
datasets. For this purpose, spatial correlations between the first six REOFs of the reference dataset CRU and all REOFs
of CCLM and GPCC are calculated to identify the best matching patterns among them. A two-sided t-test was conducted
to address the correlations' significance. The retained first six CRU REOFs account for around 75% of the total explained
variance in each season and eventually in each data set. For the regions definition, the 80th (75th) percentile of the
precipitation (temperature) REOF loadings is calculated. For the "paired" REOFs, six regions are defined by those grid
points that exceed the 80th (75th) percentiles of precipitation (temperature). Precipitation and temperature differences are
then calculated with respect to the reference CRU data set for the period 1980-2018 and each region. Taylor diagrams
containing the Pearson correlation coefficient, the root-mean-square error (RMSE) error, and the standard deviation
against the reference data CRU are then prepared for each region for the validation and skill assessment of the COSMO-
CLM simulations.

**2.3.3 Mean Climate Conditions: Pre-industrial and first millennium BCE**

The mean climate conditions between the first millennium BCE and the pre-industrial times are investigated by addressing
differences in mean values and standard deviations together with their statistical significance with a student's t-test at
each grid point at the 95% confidence level. By examining temperature variations during these periods, we gain valuable
insight into differences at the beginning and end of the simulation period. In order to investigate the relevant atmospheric
circulation in the two periods, the non-rotated temperature and precipitation PCs were linearly regressed onto the global
sea level pressure (SLP) anomalies from the MPI-ESM-P simulation "Mythos" to estimate the local regression
coefficients for the SLP field, see Eq. (3):
$$PC(j) = \beta_0(j,k) + \beta_1(j,k) * SLP(k) + \varepsilon(j,k) \tag{3}$$

The index $k$ represents the grid-point index covering the geographical domain. $PC(j)$ is the $jth$ non-rotated principal
component and $\beta_1(j,k)$ presents the regression coefficients for the $jth$ component on grid point $k$, and $SLP(k)$ is the SLP
time series of grid point $k$. $\beta_0(j,k)$ is the intercept of the regression and $\varepsilon(j,k)$ as the noise component of grid point $k$
for the $jth$ non-rotated principal component. In our case $j = 1,2,3$, i.e., the leading three non-rotated PCs. Please note
that in this study, only the regression coefficients $\beta_1(j,k)$ are used to plot the regression map.



**3 Results and Discussion**
**3.1 Evaluation of the CCLM output**
**3.1.1 Precipitation – EM**
The spatial correlations between the seasonal CRU, CCLM and GPCC REOFs for the period 1980-2018 are presented in
Table 1. The two-sided t-test at the 95% confidence level showed that all correlations are significant. CRU and CCLM
REOFs seem to agree well with the highest significant correlation of 0.94 for winter (DJF) and summer (JJA). Lower
correlations, however, characterize mainly REOFs of lower explained variance.
**Table 1: Spatial correlation of the CCLM with CRU and GPCC precipitation REOFs for winter (DJF) and summer (JJA) over**
**the EM. All values are significant at the 95% significance level. Numbers in parentheses give the corresponding REOFs of each**
**data set.**

| Winter (December to February) | | | | | |
|---|---|---|---|---|---|
| $r_{CRU-CCLM}$ | 0.70 (1 & 4) | 0.85 (2 & 1) | 0.91 (3 & 2) | 0.74 (4 & 8) | 0.74 (5 & 7) | 0.88 (6 & 5) |
| $r_{GPCC-CCLM}$ | 0.62 (1 & 4) | 0.83 (2 & 1) | 0.92 (3 & 2) | 0.66 (4 & 6) | 0.85 (5 & 5) | 0.65 (6 & 4) |
| Summer (June to August) | | | | | |
| $r_{CRU-CCLM}$ | 0.94 (1 & 1) | 0.86 (2 & 3) | 0.67 (3 & 2) | 0.67 (4 & 5) | 0.40 (5 & 5) | 0.74 (6 & 11) |
| $r_{GPCC-CCLM}$ | 0.92 (1 & 1) | 0.79 (2 & 3) | 0.69 (3 & 11) | 0.71 (4 & 4) | 0.60 (5 & 2) | 0.33 (6 & 2) |


The derived winter and summer precipitation regions are shown in Figure 2. Each map is based on the paired REOFs of
each data set with CRU, as shown in Table 1, and the regions are numbered according to the CRU REOFs ranking. For
example, GPCC winter region 5 corresponds to GPCC REOF11 as the counterpart to CRU REOF5. The explained
variance of each REOF is presented along the regions. The three precipitation data sets show a stronger agreement during
winter over the EM. Some regions show a very good agreement between the data sets, such as the winter precipitation
region 2 of CCLM (REOF1) and CRU (REOF2) over the Balkans. Other areas, such as the CCLM region 4 in winter do
not agree well with CRU and GPCC, and the CCLM regions agree less well in the summer with both CRU and GPCC,
especially along the Aegean Sea and the Black Sea coasts. In general, summer precipitation over the EM is mostly
convective and thus very local, leading to a lack of agreement between the observational data sets and the simulation. The
strong bias shown in Figure 3 might be due to an underestimation of its strength and cloud cover (Bucchignani et al.,

264 2016).

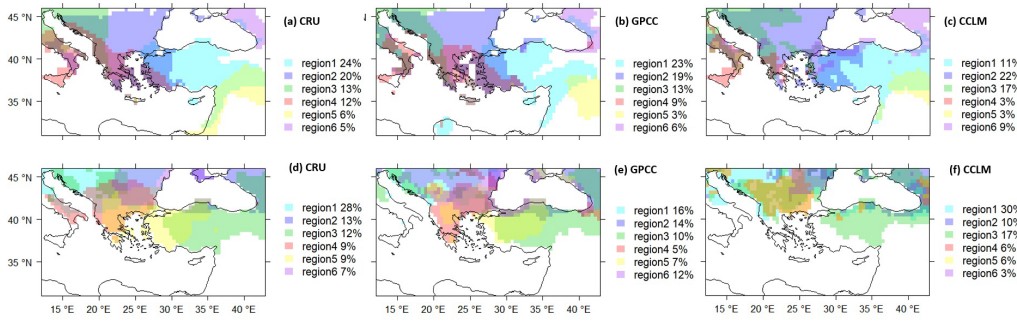


**Figure 2: Regionalization of precipitation over the EM in winter (DJF, upper panel) and summer (JJA, lower panel) for CRU**
**(a, d), GPCC (b, e) and CCLM (c, f). The total explained variance of the corresponding REOF is shown in the legend.**





The mean regional total precipitation differences of the six regions for GPCC and CCLM with respect to the 1980-2018
mean total CRU winter and summer precipitation are shown in Figure 3. The simulated seasonal variability of
precipitation agrees well with the two observational data sets, although the systematic underestimation of CCLM in
summer is noticeable. Very good agreement is visible for winter precipitation in regions 2 and 3 around the west and
south coast of the Black Sea area. The simulated mean total precipitation differences with respect to CRU for the six
regions show a correlation higher than 0.7 and comparable standard deviation for both seasons and data sets, implying
that the CCLM is performing well in representing the precipitation variability in the EM region (see Figure B1).

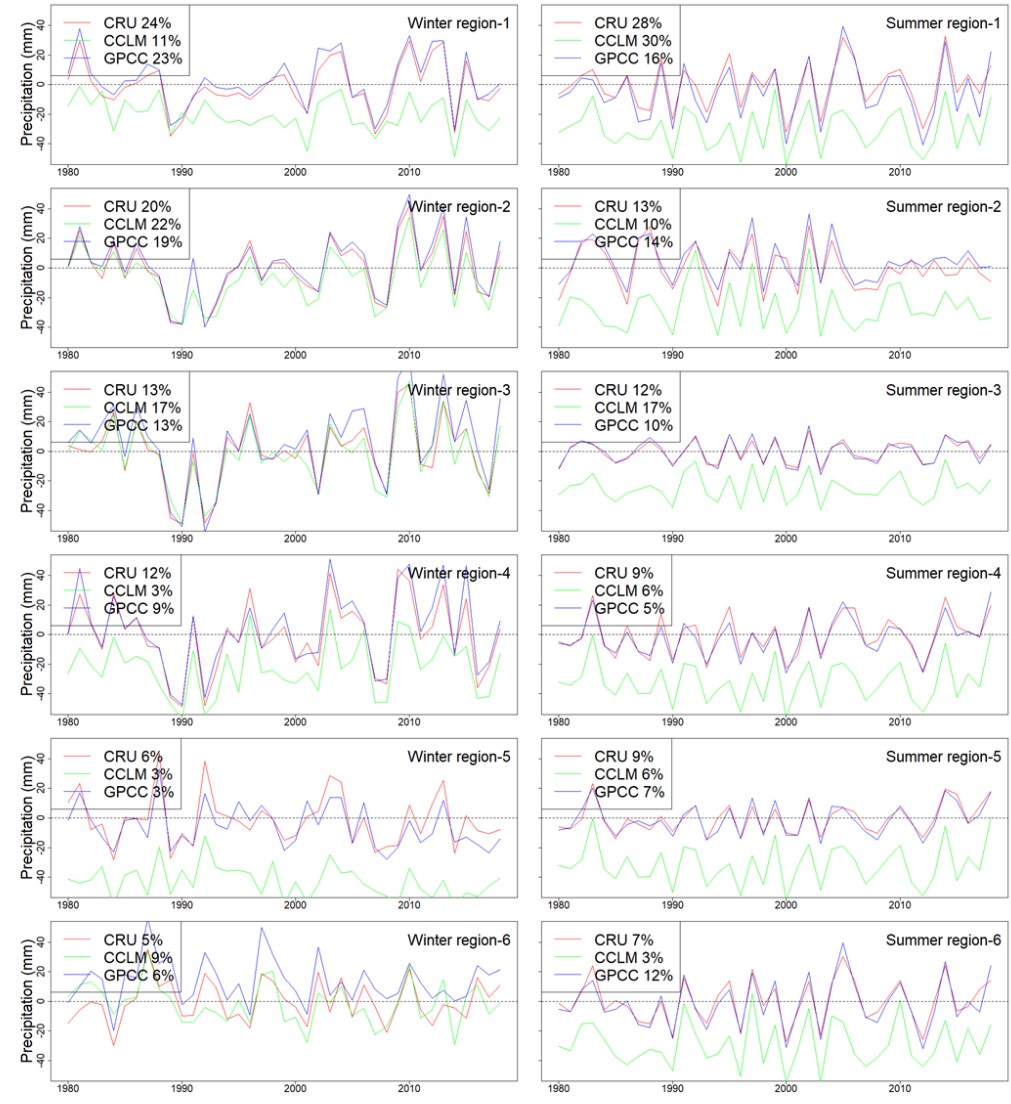

**Figure 3: Seasonal mean total precipitation differences of EM regions with respect to the 1980-2018 mean total CRU winter**
**(DJF) and summer (JJA) precipitation.**



### 3.1.2 Precipitation – NR

The EM and NR are characterized by significantly different precipitation regimes, and we have thus separated the two areas for the analysis. The spatial correlation among the REOFs of the three data sets is calculated to determine the associated regions with respect to CRU, as presented in Table 2. The correlation of REOFs for precipitation in NR is for some pairs significantly lower compared to the correlations of EM. The CCLM does not perform as well in simulating precipitation over NR, which may be related to the effect of the Hadley circulation, strong convection in the ITCZ combined with subtropical advection of hot, dry air linked with the near-surface trade winds (Adam et al., 2016). The poorer performance of CCLM over the African region may further be connected to an inaccurate representation of cloud coverage, particularly along the ITCZ (Sørland et al., 2021). Finally, the NR station time series that enter the gridded CRU and GPCC data sets are sparse, which may also reduce their quality (Harris et al., 2020; Schneider et al., 2022).

**Table 2: Spatial correlation of the CCLM with CRU and GPCC precipitation REOFs for winter (DJF) and summer (JJA) over the NR. All values are statistically significant at the 95% significance level. Numbers in parentheses give the corresponding REOFs of each data set.**

| | | | Winter (December, January, February) | | | |
|---|---|---|---|---|---|---|
| $r_{CRU-CCLM}$ | 0.79 (1 & 2) | 0.71 (2 & 1) | 0.29 (3 & 9) | 0.50 (4 & 5) | 0.74 (5 & 1) | 0.50 (6 & 4) |
| $r_{GPCC-CCLM}$ | 0.77 (1 & 2) | 0.77 (2 & 1) | 0.60 (3 & 3) | 0.62 (4 & 5) | 0.52 (5 & 6) | 0.21 (6 & 4) |
| | | | Summer (June, July, August) | | | |
| $r_{CRU-CCLM}$ | 0.64 (1 & 4) | 0.56 (2 & 1) | 0.37 (3 & 8) | 0.63 (4 & 4) | 0.40 (5 & 13) | 0.26 (6 & 7) |
| $r_{GPCC-CCLM}$ | 0.55 (1 & 4) | 0.60 (2 & 4) | 0.48 (3 & 5) | 0.32 (4 & 13) | 0.18 (5 & 8) | 0.41 (6 & 3) |

Figure 4 shows the corresponding regions based on the REOFs for each of the three data sets (CRU, GPCC, and CCLM), the explained variance of each REOFs is shown for each region. Precipitation in NR is mainly concentrated in the southern part of the area, and south of the Sahara desert. In addition, the regions of CCLM over NR exhibit a less homogeneous pattern compared to the other two data sets. This may be also related to the parameterization of CCLM which is better optimized for Europe than other areas, especially compared to complex and completely different geographical regions such as NR (Sørland et al., 2021). In the summer, the regions shift farther north compared to winter, which is related to rainfall variability due to the seasonal shift of the ITCZ (Nicholson, 2018) and the monsoon rain band (Dieng et al., 2016). The explained variance varies more between winter and summer compared to EM. For instance, the selected six leading REOFS of CCLM explain 70 % of the variance in winter but only 50 % in summer.



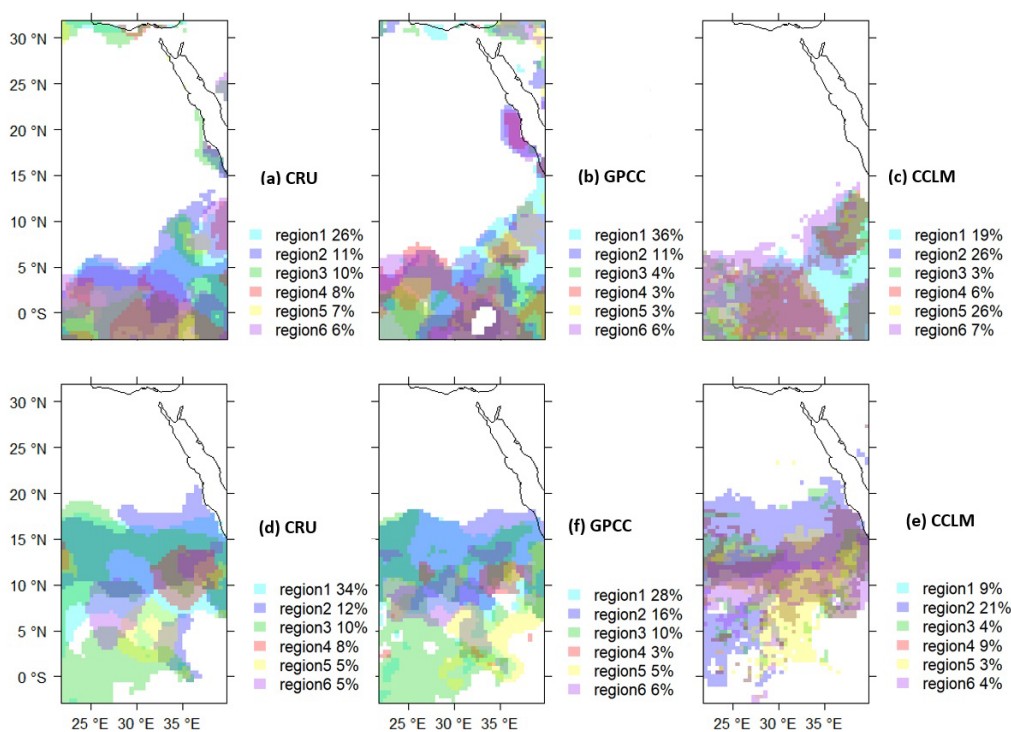

**Figure 4: Regionalization of precipitation over the NR in winter (DJF, upper panel) and summer (JJA, lower panel) for CRU (a, d), GPCC (b, e) and CCLM (c, f). The total explained variance of the corresponding REOF is shown in the legend.**

The seasonal precipitation differences, with respect to CRU (1980-2018), are presented for the six identified regions of NR (Fig. 5). Differences between the simulated and observed winter precipitation are smaller than those for EM, and all wet and dry extremes are overestimated by CCLM. In summer, there is a lack of agreement between the simulated and observed precipitation, in most cases, while the two observational data sets fail to agree as well. This is visible in the low correlations between the simulated and observational REOFs. As for the EM region, the Taylor Diagram illustrates the correlation and standard deviation of the seasonal mean total precipitation differences across the six regions in NR (Fig. B2). Compared to the EM region, the simulated precipitation in NR exhibits larger biases in terms of standard deviation compared to CRU (Fig. B2).



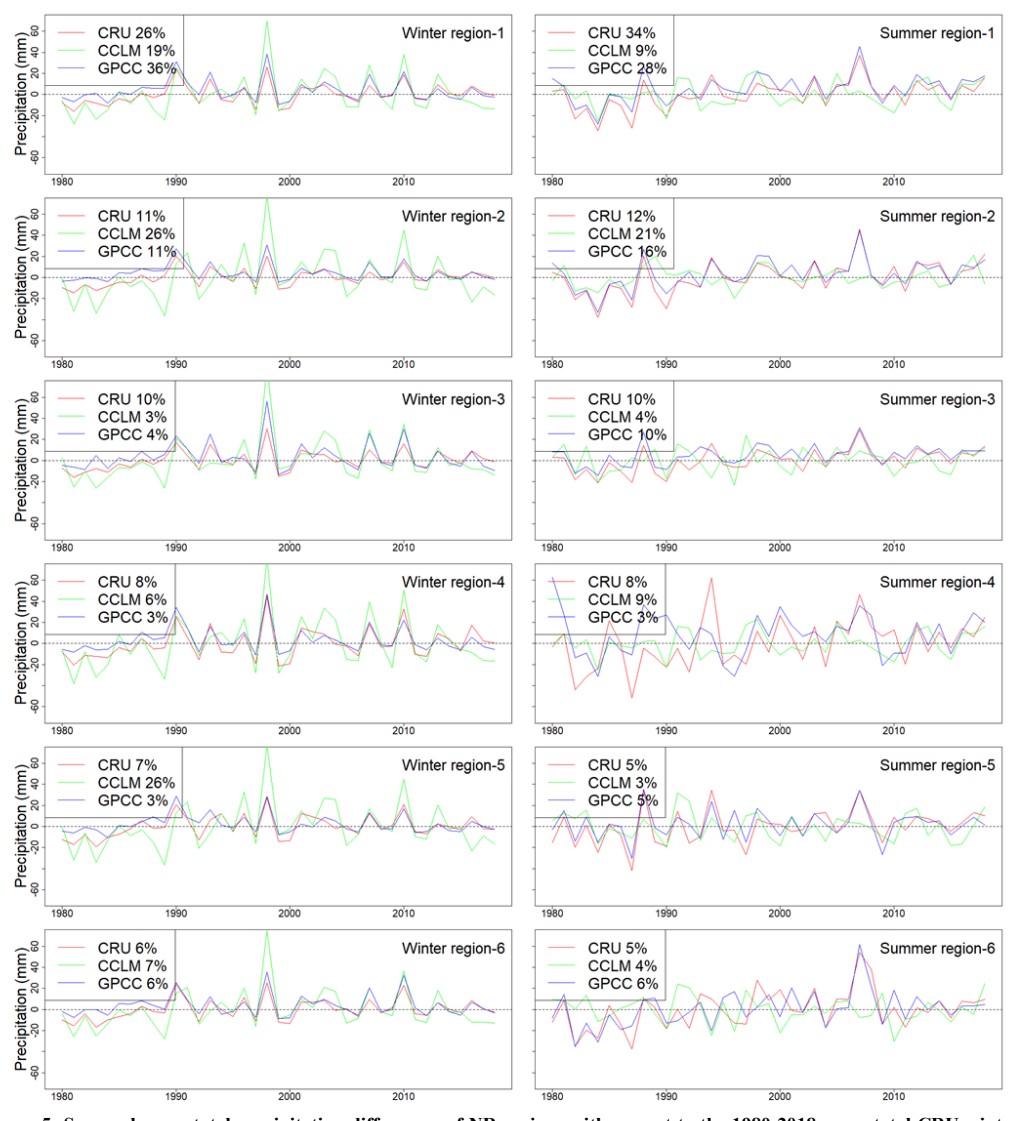

**Figure 5: Seasonal mean total precipitation differences of NR regions with respect to the 1980-2018 mean total CRU winter (DJF) and summer (JJA) precipitation.**

### 3.1.3 Temperature - EMNR

The temperature analysis encompasses the Eastern Mediterranean and Nile River Basin (EMNR). In Table A1 are presented the correlation coefficients between the three datasets REOFs (CRU-CCLM, CRU-ERA). The simulated temperature is highly correlated with both CRU and ERA-Interim data in winter and in summer, supporting the good performance of the RCM for this variable.

Figure 6 shows the regions based on the temperature REOFs of the three data sets and the explained variance of the REOFs for each region. The regions have a good agreement in space, which indicates as well the good agreement of the temperature variance of each region. Overall, the regions in all three data sets are spatially coherent regardless of the



season. The mean regional temperature differences of the six regions for GPCC and CCLM with respect to the 1980-2018
mean total CRU winter (DJF) and mean summer (JJA) temperature are shown in Figure B3. A clear agreement between
the interannual variability of the CCLM simulations and the observations/reanalysis is observed, however, differences are
found mainly in the maximum and minimum temperatures for specific regions and seasons. In region 1, CCLM
underestimates temperature both in winter and summer. Similarly, the simulated temperature is clearly lower in region 5
during winter and higher in region 6 during summer compared to the observational data sets. Such deviations from
observations in the simulated temperature may be connected with differences in simulated cloud cover and other variables
in the equatorial area affected by the ITCZ (Sørland et al., 2021).
An overall strong correlation between the CCLM seasonal mean temperature and the observational/reanalysis data sets is
further illustrated in the Taylor diagrams (Fig. B4). The over/underestimation of temperature in specific regions is
observed in Figure B3, and those regions also show lower correlation to CRU data (Fig. B4). During winter, a better
agreement is observed between simulated and observational temperatures compared to summer.

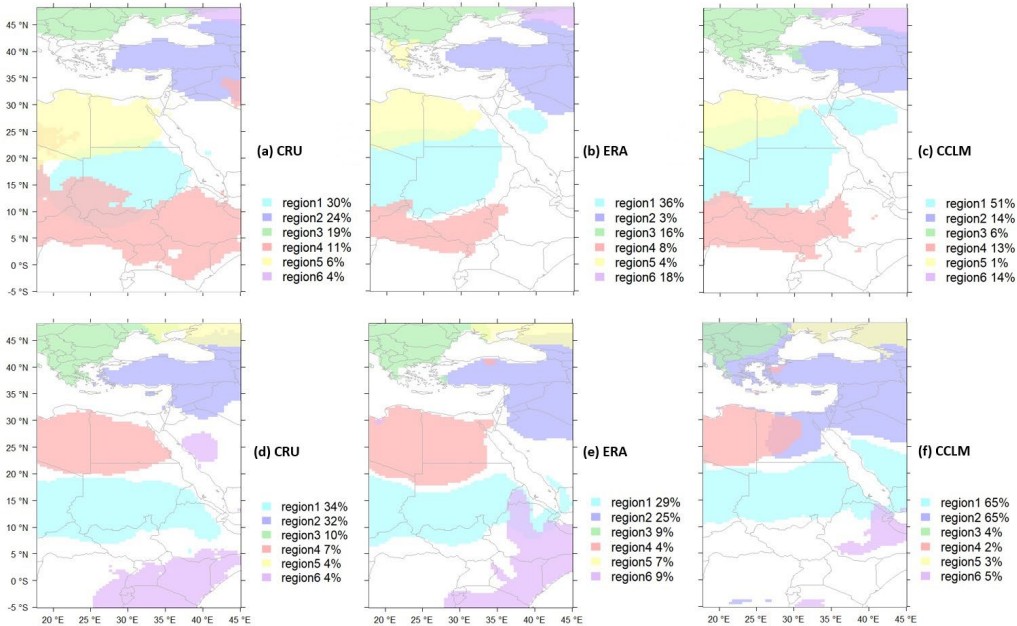


**Figure 6: Regionalization of temperature over the EMNR in winter (DJF, upper panel) and summer (JJA, lower panel) for**
**CRU (a, d), GPCC (b, e) and CCLM (c, f). The total explained variance of the corresponding REOF is shown in the legend.**
**3.2 Early Roman Period and Pre-Industrial climates: similarities and differences**
In this section, we analyze the CCLM simulation for two periods in the past, namely the PI (1800-1850 CE) and ERP
(400-362 BCE) periods. The selection of the two periods is based on two reasons: Firstly, both periods experienced a
series of volcanic eruptions as reflected in the reconstructions by Toohey and Sigl, (2017), and secondly, the two periods
represent the earliest and latest periods of our transient simulation (from 500 BCE to 1850 CE).
Figure 7 presents the precipitation and temperature annual cycle of the three regions (EM, NR and EMNR) and the
selected two periods. The absolute monthly mean total precipitation and monthly mean temperature are displayed on the
left y-axis, while the right y-axis represents the differences between the ERP and PI of each month for precipitation and



temperature, respectively. Similar mean climate conditions characterize the two periods over the study area. The
precipitation and temperature annual cycles of the two periods are similar, with slight differences for precipitation in the
NR, also reflected in EMNR. In the EM region, the largest differences in monthly precipitation of approximately 5 mm
occur in April and in May, whereas in the NR region, the most notable differences of 15 mm occur in September. The
ERP temperatures are approximately 1 °C warmer in September compared to the PI period in the EM and up to 1 °C
cooler in May over the three study areas. Similar findings were obtained when comparing CCLM simulations with and
without orbital forcing. The orbital forcing leads to increased autumn temperatures while causing a decrease in winter
and spring temperatures (Hartmann et al., submitted). Nevertheless, it is important to note that Figure 7 represents the
spatial mean of each region, and therefore, spatial variations may be averaged out.

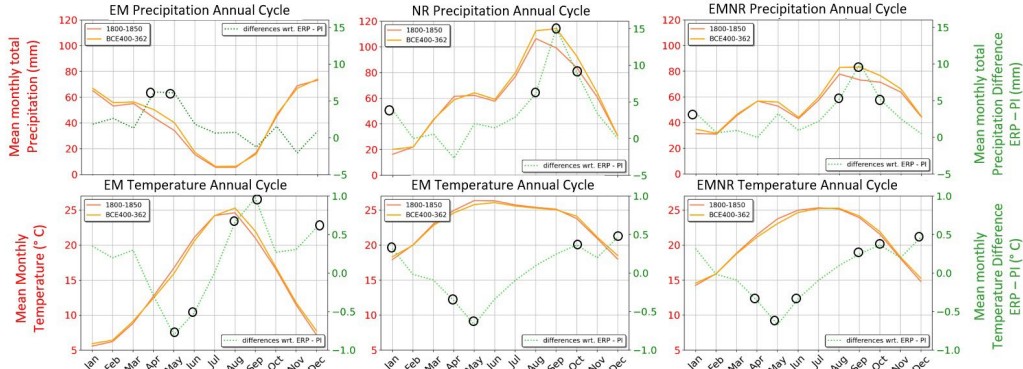

**Figure 7: Precipitation and temperature annual cycle of the PI and ERP periods for EM, NR and EMNR. Monthly mean total precipitation and monthly mean temperature are presented on the left y-axis. Differences between ERP and PI (ERP minus PI) are shown on the right y-axis. Significant differences at a 95% confidence level according to a two-sided student t-test are marked with a black circle.**

To explore the spatial differences between the two selected periods (ERP and PI) and the three regions (EM, NR, EMNR),
we calculated the mean and standard deviation differences (ERP minus PI) for precipitation and temperature, shown in
Figure 8. The dotted areas indicate statistically significant differences assessed with a t-test at the 95% confidence level.
There are notable differences in precipitation over the EM region (Fig. 8, upper panel, red rectangle), in both winter and
summer. Compared to the PI period, the ERP winters experienced lower precipitation along the northern and eastern
coasts of the Eastern Mediterranean Sea, and increased rainfall over the Balkans and the southern coast of the Black Sea.
In addition, the ERP winter season shows less variability over the entire Greek peninsula and more variability over the
northern Balkans and southern Black Sea. Summers during the ERP period are relatively wetter across the entire EM
region compared to the PI period, with statistically significant differences in some parts of the Balkans. The variability is
higher in most parts of the EM region during the ERP summer period, except the eastern coast of the Black Sea. However,
in both winter and summer, the mean differences are not statistically significant at the 95% level over most of the EM
area.
The blue rectangles in the upper panel of Figure 8 show the precipitation differences between the two study periods for
the NR region. During the winter, most of the NR region experienced drier conditions during the ERP compared to the
PI. However, significantly wetter winters (about 20 mm) are found for the southwestern part of the NR region. In summer,
the difference in rainfall between the two periods is more pronounced in the NR compared to the EM, with more areas
experiencing statistically significant wetter conditions during the ERP than in the PI, and drier conditions over Southern
Sudan and the Ethiopian Highlands. It is noteworthy that the standard deviation in the summer shows considerable



differences across the NR region reflecting the underlying hydrological conditions and local-scale tropical convective
activity.
The lower panel of Figure 8 shows the temperature differences for the three regions (EM, NR, EMNR). Accordingly, the
ERP winters are warmer compared to the PI period, particularly over the EM and the northern parts of NR. Among the
three studied regions, the difference in summer temperatures between the two periods is relatively small, within 0.5 °C.
On the other hand, the standard deviation of temperature during the two periods shows distinct differences between
summer and winter. The ERP winter temperature is overall less variable compared to the PI period, whereas the ERP
summers show higher temperature variance (around 0.7 °C), especially over the EM region and the northern NR region
compared to PI.

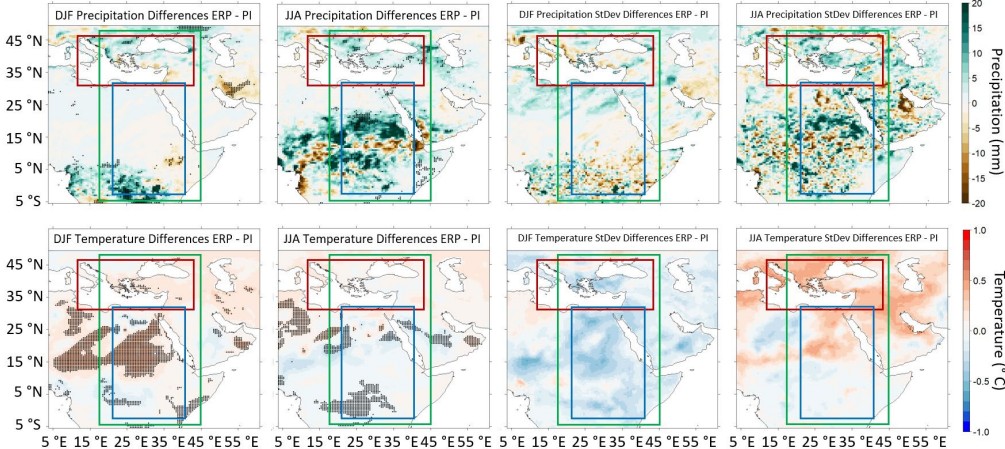


**Figure 8: Seasonal differences (ERP minus PI) for winter (DJF) and summer (JJA) mean and standard deviation (StDev) of
total precipitation over EM (red rectangle), NR (blue rectangle) and 2m air temperature over EMNR (green rectangle). Dotted
areas indicate statistical significance at the 95% confidence level according to the local student's t-test.**

**3.3 Connection to atmospheric circulation**
To establish the link between the large-scale atmospheric circulation and local climate in the ERP and PI periods, we
explore in this section the connections between SLP and seasonal precipitation and temperature. We present results for
winter precipitation over EM and summer temperature over EMNR and we restrict from using the precipitation data over
the NR region part due to the limitations of CCLM to accurately simulate precipitation over the area. By examining the
leading non-rotated EOFs, we gain insight into the dominant patterns of variability in winter precipitation and summer
temperature within these two regions.
Figure 9 displays the three leading EOF patterns for the PI (a, b, c) and ERP (d, e, f) periods with the corresponding
explained variance. For both periods, they show similar patterns with 75% cumulative total explained variance. EOF1
resembles the mean winter precipitation spatial distribution with highest precipitation anomalies along the western coasts
of the peninsulas, indicating the influence of the westerly circulation and the land-sea interaction. The second EOF
displays a dipole pattern with negative precipitation anomalies over the western and positive precipitation anomalies over
the eastern part of EM, likely connected with the direct impact of the large-scale circulation. The higher index EOF3 is
characterized by a weaker dipole with distinct lower precipitation along the eastern coasts of the Black Sea and higher



precipitation in the southern Mediterranean coastal areas. Smaller differences are observed in the total explained variance
of each EOF of the two periods. This change in the relative amount of variance represented by the individual EOFs may
indicate changes in the spatial structure of precipitation variability between the ERP and PI periods.

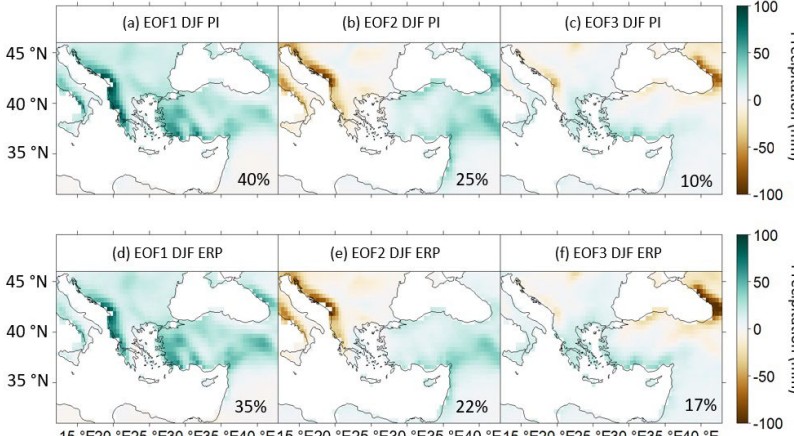


**Figure 9: Patterns of the first three non-rotated EOFs of winter (DJF) precipitation for PI (1800-1850 CE; a, b, c) and ERP**
**(400-362 BCE; d, e, f) and total explained variance for each EOF.**
In Figure 10, the regression maps (i.e., the regression coefficients $\beta_1(j, k)$ according to Eq. (3)) are shown, between the
three precipitation principal components ($PC(j)$) and the SLP fields from the driving MPI-ESM-P "Mythos" for the same
periods. All regression maps show very similar spatial patterns, albeit with varying amplitudes between the two periods.
This change in the amplitude of the regression coefficients may indicate variations in the robustness and strength of the
underlying relationships between the precipitation PCs and the large-scale SLP fields.
The EOF1 positive winter precipitation anomalies along the western coasts of the peninsulas of EM (Fig. 9a, d) are
connected with lower pressure over the Mediterranean (Fig. 10a, d) which is related to the higher frequency of cyclones
from the Gulf of Genoa or of Atlantic origin moving eastward, which, together with the orographic lifting, lead to high
amounts of precipitation over these areas. The dipole structure of winter precipitation EOF2 (Fig. 9b, e) with drier
conditions over western EM and wetter winters over its eastern part is linked to a statistically significant SLP dipole (or
tripole for ERP, Fig. 10b, e) with stable anomalous anticyclonic conditions (SLP positive anomalies) over the Eastern
Atlantic and western Europe and an anomalous trough with negative SLP anomalies over the east. The dry conditions
over the eastern coasts of the Black Sea in EOF3 (Fig. 9c, f) are connected with the prevailing anomalous high pressure
centered north of the Caspian Sea and extended over Eurasia (Fig. 10c, f). The ERP and PI periods represent a similar set
of regression patterns with different intensity for winter with respect to the influence of large-scale circulation.





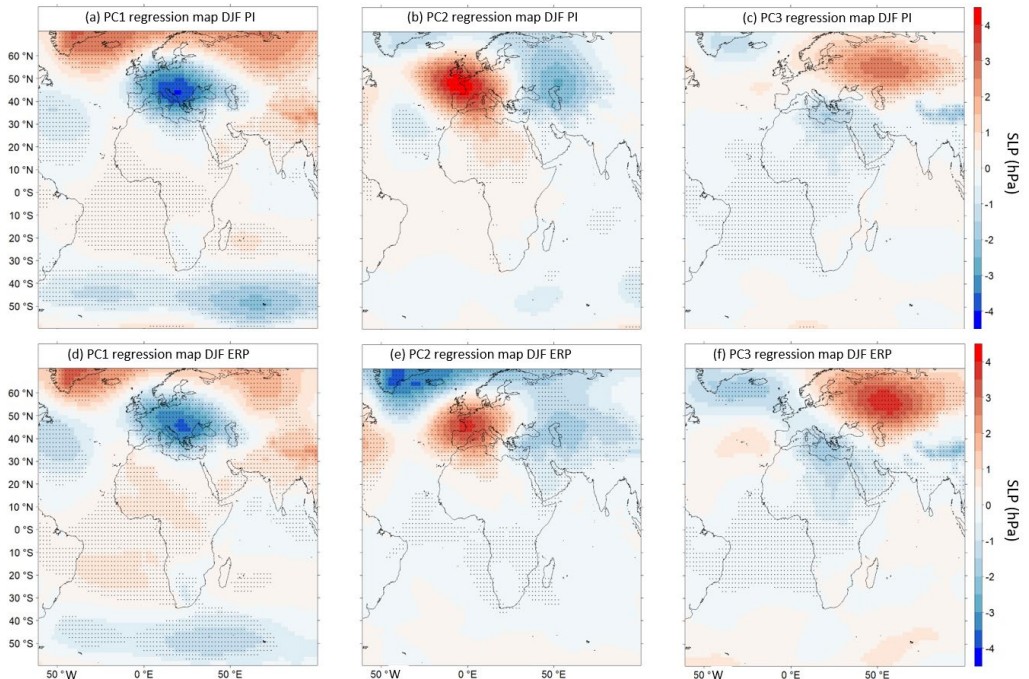

**Figure 10: Regression maps between the PCs of the first three non-rotated winter (DJF) precipitation and SLP from the Mythos simulation for the PI (a, b, c) and ERP (d, e, f) periods. Statistical significance at the 95 % confidence level is denoted with dotted areas.**

Figure 11 shows the leading three EOF patterns for summer 2m air temperatures over the EMNR region during the PI (a, b, c) and ERP (d, e, f) periods that together represent nearly 85% of the total variance. Notably, the first EOF of the PI period explains nearly half of the variance, while the ERP period exhibits a lower but still high contribution of 40%. The EOF patterns show a high degree of similarity between the PI and ERP periods. The EOF1, as depicted in Figure 11 (a and d), shows a clear temperature dipole pattern, characterized by distinctive negative temperature anomalies over the African Sahel region (likely associated with ITCZ-related variability) and pronounced positive temperature anomalies over EM, extending to the Middle East and Egypt. This dipole structure indicates the presence of an apparent teleconnection between the Mediterranean Sea /Northern Africa and the Sahel region, resembling a temperature sea-saw pattern. The second EOF air temperature patterns (Fig. 11b, e) for the PI and ERP periods show positive anomalies throughout the entire EMNR area, while an intensified positive temperature anomalies signal is shown between 10 to 20 degrees north (Sahel). This pattern indicates similar variations of the two periods over the entire region that are uncorrelated to the temperature sea-saw depicted in EOF1. The EOF3 (Fig. 11c, f) shows weak negative temperature anomalies over the African Sahel region shifted slightly to the north, compared to EOF1, with negative anomalies extending over the Middle East and weak positive anomalies over the Balkans. This pattern shows lower values in the amount of explained variance.





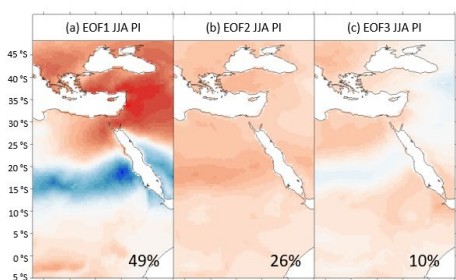
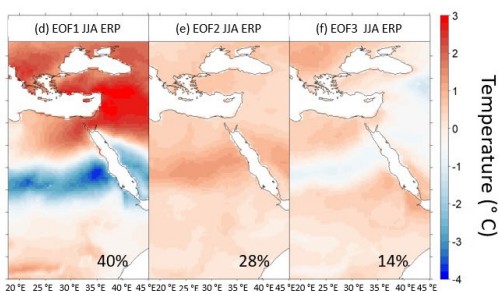

**Figure 11: Patterns of the first three non-rotated EOFs of summer (JJA) temperature for PI (1800-1850 CE; a, b, c) and ERP (400-362 BCE; d, e, f) and total explained variance for each EOF.**

The strong positive temperature anomalies over the northern and eastern part of the area and the belt of negative anomalies over the subtropical region in EOF1 is reflected in the SLP regression map (Fig. 12a, d). The EOF1 temperature pattern is related to negative SLP anomalies over the northern North Atlantic and positive anomalies over the central North Atlantic, potentially connected with an intensified subtropical anticyclone, as well as negative anomalies over the extended study area, which may indicate the influence of the Persian trough extend from the Asian Monsoon trough (Lelieveld et al., 2012). Positive SLP anomalies are located over the belt of negative anomalies over the subtropical region. The EOF2 positive temperature anomalies over the entire EMNR area are connected to negative SLP anomalies over the region (Fig 12b, e). The higher temperatures over the area may be connected with an intensified Sahara heat low, which is an area of low surface pressure as a response high low-level temperatures (Lavaysse et al., 2009; Messager et al., 2010). The third EOF of the two periods (Fig. 11c, f) explains a smaller amount of variance and thus the interpretation of the patterns and the corresponding regression maps (Fig. 12c, f) is challenging and here not attempted.

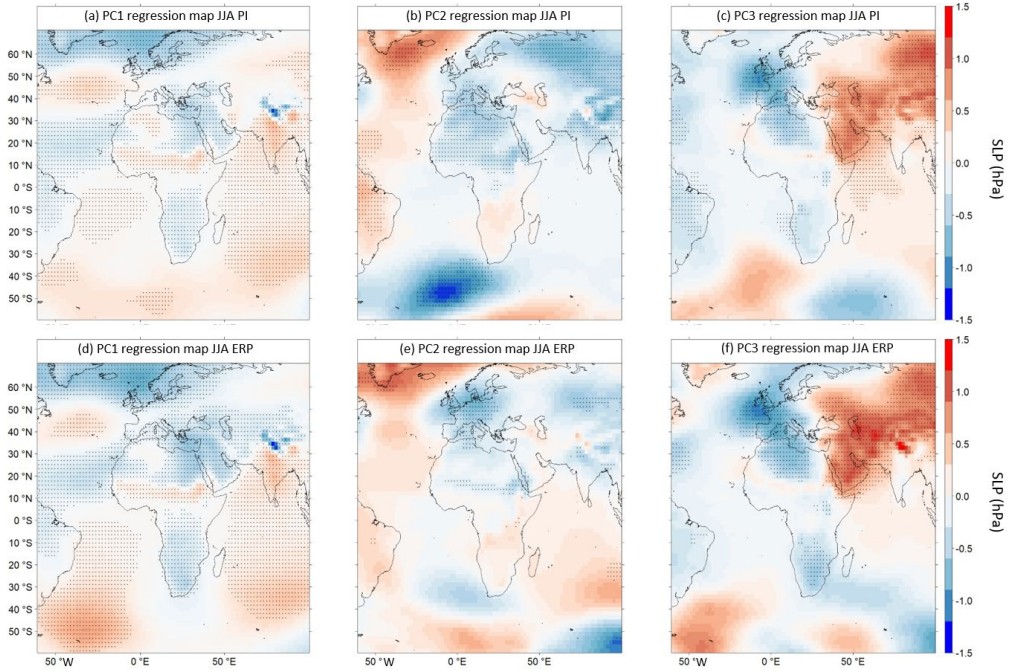


**Figure 12: Regression maps between the PCs of the first three non-rotated summer (JJA) temperature and SLP from the**
**Mythos simulation for the PI (a, b, c) and ERP (d, e, f) periods. Statistical significance at the 95 % confidence level is denoted**
**with dotted areas.**
**4 Conclusions**
In this study, we presented the first fully forced CCLM adapted for paleoclimatic applications including volcanic, solar,
land-use, greenhouse gases, and orbital forcings with a spatial resolution of 0.44° over the extended area of the Eastern
Mediterranean and the Nile River Basin. We evaluated the performance of the model in the present time (1980-2018) and
compared the simulated climate for Early Roman Period, ERP (400-362 BCE) and the pre-industrial, PI (1800-1850 CE)
period.
Our study demonstrates that the fully forced paleo CCLM can simulate reasonably the climate within the selected region.
In general, the model exhibits better performance in simulating the seasonal air temperature compared to precipitation.
Especially within the Nile River domain, the simulated precipitation is less accurate, possibly due to the model's
limitations in representing convective processes linked to the ITCZ. Another important factor relates to the sparse network
of observational data over those areas, further complicating a proper comparison between simulated and observed
precipitation. In contrast, the model performs better in capturing precipitation patterns over the eastern Mediterranean.
The drying bias that occurs in the EM (especially in summer) may be related to the model's underestimation of total cloud
cover. Moreover, the model tends to underestimate temperatures in northern Africa (around 10° - 32° N) during winter
and the Sahel (10° - 20° N) during summer. During summer, temperatures in North Africa and the EM are showing a
warm bias.
In terms of the annual cycle of precipitation and temperature, the average climatic conditions were comparable between
the two periods, but the climatic variability varied across the study area. The ERP over the EM was generally wetter than



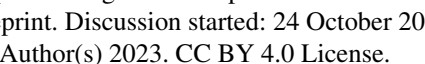



the PI in both summer and winter, with greater variability in summer. In the NR region, there were no statistically
significant differences during the ERP winter compared to the PI, while summers during the ERP were predominantly
wetter and reflected a larger variability compared to the PI period. However, we have found that the CCLM has limitations
in capturing a realistic rainfall pattern over the NR region.
The links between the regional precipitation and temperature patterns and large-scale features were investigated using a
linear regression approach between the principal components of summer temperature and winter precipitation onto the
sea level pressure. The according results indicate consistent associations between precipitation/temperature EOF patterns
and corresponding SLP anomalies during both periods. The major winter precipitation patterns over EM shown in the
principle component analysis are found to be related to the cyclones/anticyclones of the surrounding area (such as the
Eastern Atlantic, the Western Europe) together with the orographical lifting. While, the NR region is located under the
ITCZ and is subject to various circulation mechanisms, which poses a challenge for the model to accurately simulate the
local climate and its variability.
Future studies may involve adjusting physical parameters such as albedo, soil layers and aerosols to improve the
performance of COSMO-CLM in capturing precipitation patterns in the NR region. Detailed investigations of the linkages
between temperature and specific large-scale circulations in the EMNR region are also needed to provide a comprehensive
understanding of the teleconnections in the study area. For paleoclimate research, comparing modeled data with proxy
records is essential for a comprehensive understanding of climate variability and change over the past 2500 years and
improvement of the climate models. Additionally, establishing connections between climate events and historically
documented societal developments can provide insights into the role of climate change in the context of historical socio-
economic changes.
**Appendix A**
**Table A1: Spatial correlation of the CCLM and ERA with the first six CRU temperature REOFs for winter (DJF) and summer**
**(JJA) over the EMNR. All values are significant at the 95% significance level. Numbers in parentheses give the corresponding**
**REOFs of each data set.**

| Winter (December, January, February) | | | | | |
|---|---|---|---|---|---|
| $r_{CRU-CCLM}$ | 0.79 (1 &1) | 0.78 (2 & 2) | 0.88 (3 & 5) | 0.93 (4 & 3) | 0.73 (5 & 10) | 0.57 (6 & 2) |
| $r_{CRU-ERA}$ | 0.93 (1 & 1) | 0.93 (2 & 2) | 0.93 (3 & 5) | 0.94 (4 & 3) | 0.84 (5 & 4) | 0.85 (6 & 10) |
| Summer (June, July, August) | | | | | |
| $r_{CRU-CCLM}$ | -0.76 (1 & 1) | 0.69 (2 & 1) | 0.73 (3 & 3) | 0.51 (4 & 9) | 0.86 (5 & 6) | 0.75 (6 & 2) |
| $r_{CRU-ERA}$ | -0.77 (1 & 1) | 0.64 (2 & 2) | 0.78 (3 & 3) | 0.83 (4 & 2) | 0.90 (5 & 6) | 0.67 (6 & 4) |





**Appendix B**

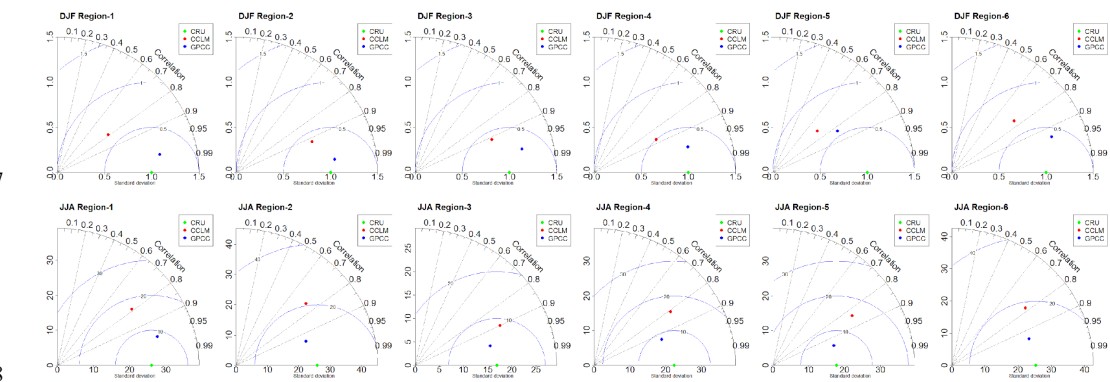

**Figure B1: Taylor diagrams of winter (DJF) and summer (JJA) precipitation differences with respect to the CRU data set for the period 1980 – 2018 for the Eastern Mediterranean regions. The green dot indicated CRU data set, therefore the correlations showing in the plot is 1, and the blue (red) dot indicate the correlations between CCLM (GPCC) with CRU data sets.**

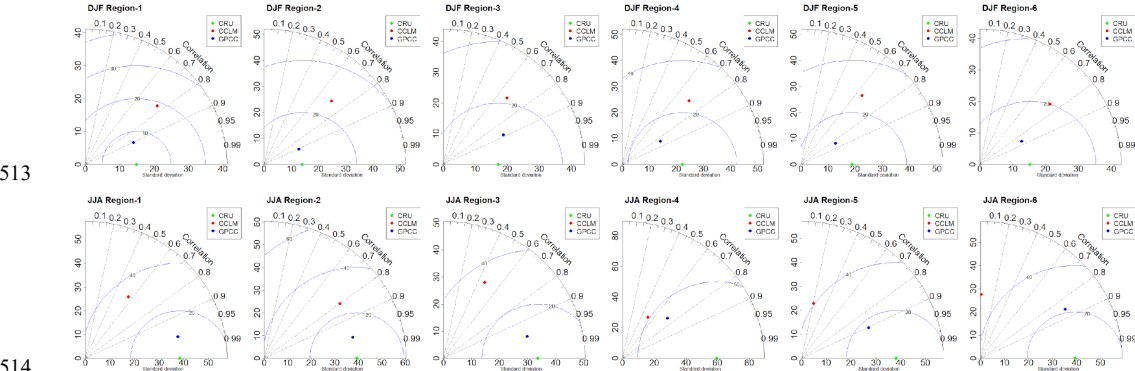

**Figure B2: Taylor diagrams of winter (DJF) and summer (JJA) precipitation differences with respect to the CRU data set for the period 1980 – 2018 for the Nile River Basin regions. The green dot indicated CRU data set, therefore the correlations showing in the plot is 1, and the blue (red) dot indicate the correlations between CCLM (GPCC) with CRU data sets.**






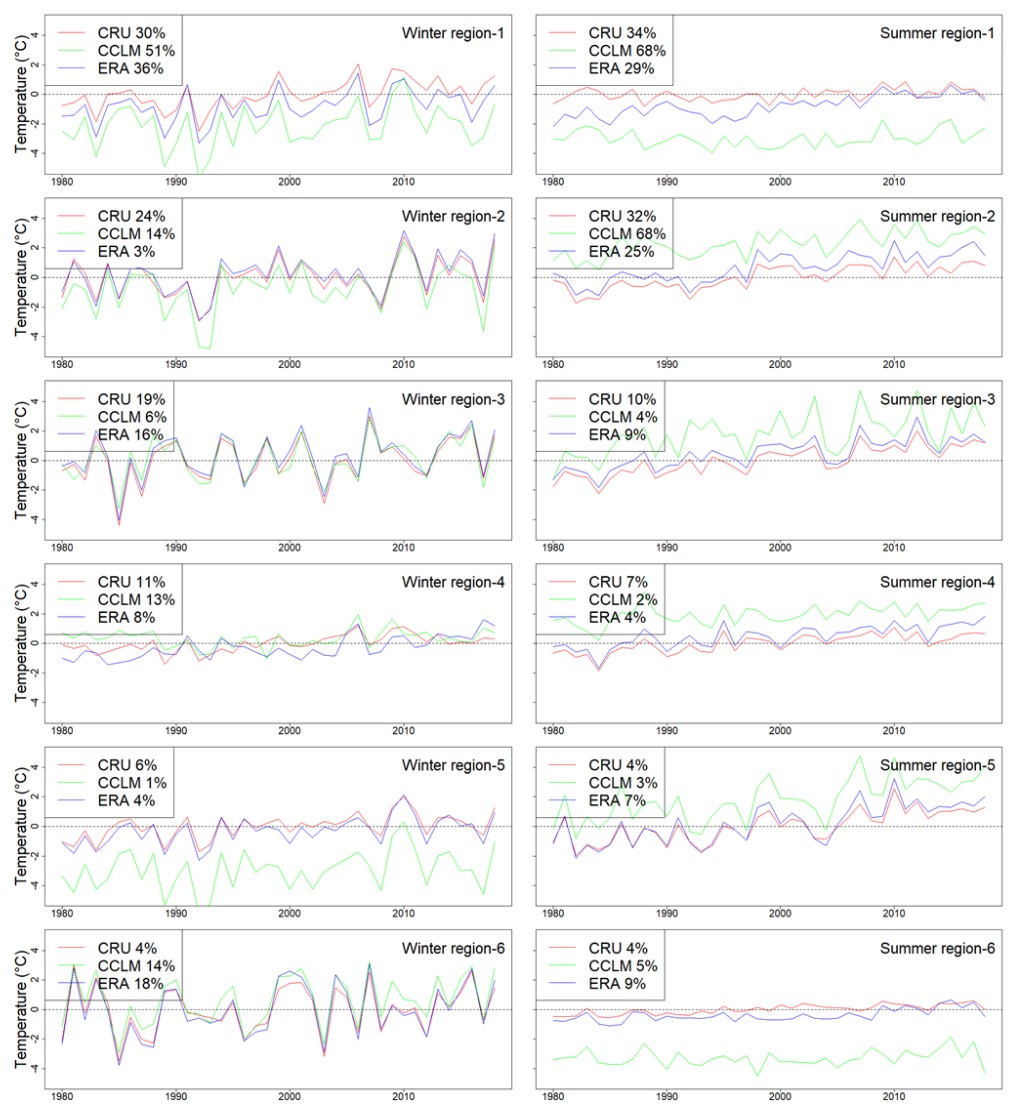

**Figure B3: Seasonal mean total temperature differences of EMNR regions with respect to the 1980-2018 mean total CRU winter (DJF) and summer (JJA) temperature.**





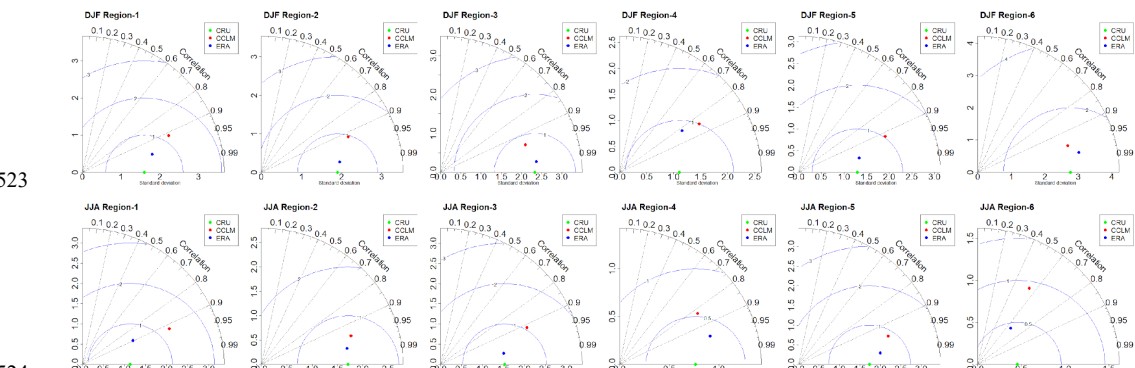



**Figure B4: Taylor diagrams of winter (DJF) and summer (JJA) precipitation differences with respect to the CRU data set for the period 1980 – 2018 for the Eastern Mediterranean/Nile River Basin regions. The green dot indicated CRU data set, therefore the correlations showing in the plot is 1, and the blue (red) dot indicate the correlations between CCLM (ERA-Interim) with CRU data sets.**

**Code availability**

The COSMO-CLM model is available and free of charge for all members of the CLM-Community via their website https://www.clmcommunity.eu/. The user must either be a member of the CLM-Community or the respective institute must possess an institutional license. The changes in the original source code for implementing the external forcings are described in detail in Hartmann et al., (submitted).

**Data availability**

The simulated data by COSMO-CLM in this paper is stored at the Deutsches Klimarechenzentrum (DKRZ).

**Competing interests**

The authors declare that they have no conflict of interest.

**Author contribution**

MZ and EH developed the model code and performed the simulations. EX, SW, MA and MZ contributed to the conceptualization of the manuscript. MZ conducted the formal analysis with the support of SW and NL. MZ prepared the manuscript. All authors followed the analysis from the beginning, provided text and edited/commented the final version of the manuscript.

**Acknowledgement**

This work used resources of the Deutsches Klimarechenzentrum (DKRZ) granted by its Scientific Steering Committee (WLA) under project number bb1201. EX acknowledges support by the Greek "National Research Network on Climate Change and its Impact" (project code 200/937). EX, NL acknowledge support by the EU Horizon 2020 Project CLINT

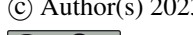



under Grant Agreement number 101003876. EX, MA acknowledge support by the German Federal Ministry of Education
and Research (BMBF) project NUKLEUS (grant number 01LR2002F).

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
