# Peer review of "The climate of the Eastern Mediterranean and the Nile River basin"

_Climate of the Past, 2023_

## Referee Comment (RC3)

**Review of "The climate of the Eastern Mediterranean and the Nile River basin 2000 years ago using the fully forced COSMO-CLM simulation" by Zhang et al. 2023**

**General Comments**

In this manuscript the authors present the results of the first fully-forced, high resolution simulation of the climate of the Eastern Mediterranean and Nile river regions over the last 2500 years using the COSMO-CLM regional climate model. The work is mainly divided in two parts. The first one, where the authors evaluate the model with several modifications necessary for the simulation of past periods, for the present-day. And the second one where they assess climate changes between two past periods of time, namely the Early Roman Period (ERP) and the Pre-Industrial (PI) period. Here they assess differences in seasonal values of precipitation and temperature, as well as their connection to changes in the atmospheric circulation.

The paper presents some interesting results and its contents fit well within the scopes of Climate of the Past. Nonetheless, I do believe that the manuscript suffers from a series of major issues that need to be properly addressed before it could be considered for publication for the journal.

First of all, the objectives of the paper are in my opinion not very well defined. The employed methods are not always clearly described, both concerning the description of the experimental design of the presented simulations as well as for the statistical methods employed in the analysis of their results. This makes the understanding of the different analysis not always straightforward. Additionally, the presented analysis can sensibly be extended, making a full use of the transient simulation and of the driving GCM data, to understand discriminated model biases, as well as possible differences in the climate of different time frames of the simulation period. The beauty of your simulations is that you have so much data from which we could really learn a lot about past climate changes of the investigated area and their drivers. I would try to make a full use of them. Finally, it would be interesting to know how the model performs in past times, performing a comparison against proxy data for the study period.

Please, find below more detailed comments on which I based my judgement.

**Major comments**

- p3, l85-87: you mention that for this area a dense network of natural archives is available covering the last 2000 years. These data should be acknowledged and used for the comparison against your model results.

- p4., l133: you need to present a summary of the model setup, particularly concerning information on how you implemented changes in the model to take into account changes in the forcing.

- section 2.3.1: since you have data, wouldn't it be better to conduct EOF analyses of seasonal anomalies over the entire simulation time period and detect possible trends? In this way you could also compare changes across different periods. I think that this analysis, considering the fact that the study presents for the first time the results of a transient simulation for the area at high resolution, would be quite interesting. Also, are the presented results sensitive to the relatively short length of the two time periods considered?

- section 2.3.2: the method you use for the clustering of the different regions according to seasonal values of precipitation and temperature is not entirely clear. This part needs to be revised and possibly extended with additional details.

- section 2.3.2: Additionally, there are many choices that seem arbitrary in your method and that need further testing: for example, l219-220, why choosing only 6 EOFs for CRU and all for the other datasets?

- section 2.3.2: Another important point: do the different regions you derived from the different datasets contain different number of grid-boxes? This is a point that needs particular attention, in particular for the conclusions you draw from Fig. 3 and Fig. 5. When you quantify the match between datasets across regions, as performed in Fig. 5 and Fig. 3, you need to consider overlapping regions with the same number of points.

- section 2.3.2: Why for the present-day you use rotated EOFs and for the investigation of past periods you use non-rotated ones?

- section 2.3.3: since you have the results, why not showing the analysis in temperature, precipitation and mean sea level pressure for the entire simulation period? I think this would give some important and interesting insights on the simulated climate of the given period and area. In any case, whenever you show the differences between the two selected periods you must use the transient results for the entire simulation period to assess whether the obtained differences are comparable to the ones of other periods or if they particularly stand out? in the latter case, you could eventually try to assess why.

- section 2.3.2 and 2.3.3: In the paper the authors do not acknowledge in any way how the outcomes and conclusions of the manuscript are subject to the series of different arbitrary criteria they applied throughout their analysis. At least some discussion is needed here, to make readers aware that some changes might occur when changing some details of the method.

- Fig3 (same for Fig. 5): Why not comparing first the mean regional climatological values for a given region between the different datasets and then comparing the anomalies of each time series calculated with respect to the corresponding mean value of each dataset. Basically, instead of calculating all the anomalies with respect to the mean value of CRU in each region, it would be more appropriate to remove from the time series of each dataset the corresponding mean for the calculation of the seasonal anomalies. In this way you would have a proper assessment of the differences in the mean in each dataset as well as in their temporal variability.

- Fig. 7: Why are you now simply comparing spatial means over the entire region? in particular, what is the need for all the previously conducted analyses on sub-regions that you performed in previous sections in this context?

- section 3.3: Alternatively, you could also consider to conduct a canonical correlation analysis between SLP and precipitation and temperature over the entire period of time.

**Minor comments**

- p1, l14: at the regional scale

- p1, l15: atmospheric dynamics

- p1, l15: for present and future climate conditions

- p1, l17: please try to better describe in the manuscript what are the teleconnections relevant for the study domain

- p1, l17: you do not develop COSMO-CLM. You rather apply a high-resolution climate model modified for its application to paleoclimate studies. Make sure in the text that some studies already applied modified versions of COSMO-CLM to paleoclimate.

- p1, l23: comparable climatic conditions between the two considered periods

- p1, l23: variability of what? please specify

- p1, l27-29: period needs reformulation

- p1, l30: shed lights into

- p2, l17: involved in what? reformulate

- p2, l59: In summer

- p3, l78-81: reference needed

- p3, l90: in all PMIP phases (and not only PMIP4) model results were compared against proxies: please reformulate this period accordingly

- p3, l94-95: Can you specify the difference between proxy records and climate reconstructions? do not climate reconstructions rely on proxy records? I think that a better choice here would be simply using proxy-based climate reconstructions.

- p3, l97-104: I miss here some discussion, also based on previous literature, on why the application of RCMs to the study of the past is relevant.

- p3, l108-111: where can I see this? a proper discussion of available proxies for the region is needed

- p3,l112: I think that here you have a good chance to introduce the work conducted in this study and the simulations performed with COSMO-CLM and MPI-ESM.

- p3, l112: the listed forcing include both internal and external forcing: please correct.

- p4, l116: here it would be appropriate to also mention some of the works of Berger about the estimation of orbital parameters for the past.

- p4, l117-118: this sentence needs reformulation.

- p4, l119: I guess just some of the Nile flooding match volcanic eruptions and not all? maybe it might be interesting to report some example?

- p4, l120: the increase in energy in the climate system is continuous only for a continuous increase in GHGs. Please modify accordingly.

- p4, l124-126: you have to introduce before in the text the GCM simulation you are using in your study, as well as the fact that you are using this for running an RCM. This is not explicitly mentioned up to this point in the introduction. See also one of my comments before.

- p4, l124-126: Also, be aware that the land cover changes are specific to the target area and study period

- p4, l127: "but those forcings are not yet fully implemented in the RCM": please be aware that many other studies with modified forcing were already performed with COSMO-CLM.

- p4, l128-130: See comments above: so far it is not clear if you will be conducting a study with COSMO-CLM. Also, note that you use sometimes COSMO-CLM and some otehrs CCLM. Following the specifications of the CLM-community, I would recommend to always use the acronym COSMO-CLM for the model throughout the manuscript

- p4, l135-136: why this association should not be possible simply using a GCM? please better clarify

- p4, l136-138: This is not shown here. I would rather frame it as the possibility to use the results for the study of extreme events on societies. Still, in this case you must make clear that you need a proper comparison against proxy data before using the model results for past times.

- p4, l152: Armstrong et al. 2019 do not use COSMO-CLM. A more appropriate reference here would be the one of Soerland et al. 2019.

- p4, l. 152: ".. have been recently performed": COSMO-CLM activities and participation to CORDEX covers more than 15 years. Please, reformulate.

- p5, l155: pleas,e be more careful in the use of extreme terms: you actually do not revolutionise COSMO-CLM since other studies have already applied at least part of the changes necessary for the application of the model to paleoclimate studies that you are considering.

- p5, l156: As already stated before, it is not sufficient to include the reference to the paper of Hartmann et al. 2023 here. You need to provide a summary of the applied changes to the model and to its configuration.

- p5, l170: for which area they apply COSMO-CLM for, in the study of Bucchignani? why using their configuration? please specify

- section 2.1: As you mention later in the text, the configuration of the model is very important for an RCM cause it is region-dependent. What was the starting setup of your model? did you use the default setup for Europe provided by the CLM-community? you did not apply any additional changes beside the ones in accordance to Bucchignani et al. 2016? Eventually, provide more context on the reason for your choices in the model setup

- p5, l177-180: can you provide here more context on why selecting the two periods 1800-1850 CE and 400-362 CE in your study? I think that, also considering your performed analysis, a more appropriate choice would be the one of two periods with the same length. Also, you have so exciting results: why not performing the analyses over the entire simulation period?

- section 2.1: please also add here the horizontal resolution of your model as well as the extent of the domain over longitudes and latitudes

- Fig1: Is the outer box the entire domain of your simulation? please specify

- p6, l186: We evaluate: evaluate is in my opinion a more appropriate choice than "validate", since we have to acknowledge that also observations are not the absolute reality

- section 2.2: please be aware that the original resolution of ERAInterim is not 0.5°. Specify if you interpolated the data onto the target grid yourself or if you simply retrieved interpolated data from the ECMWF server.

- p7, l214: corresponding instead of according

- p7, l219: maybe it would allow a better understanding of your method if adding here (or eventually in the supplements) a figure with the EOFs obtained for the CRU dataset, for both considered variables

- p7, l221-222: can you provide more details on why the CRU REOFs accounf for the 75 % of variance in in each dataset?

- p7, l222: How do you calculate the 80th percentile of the loadings? also, you calculate the 80th percentile of the loadings with respect to REOFs of CRU or for each dataset separately? How are your results sensitive to the specific arbitrary choice of 80th and 75th percentile?

- p7, l223: Not clear what you mean by "paired" REOFs. Could you please provide additional details on the pairing methodology?

- p7, l226-228: specify that you produce the Taylor diagrams also for each season. In the Taylor diagrams, do you compare results over the same region? If yes, are these the regions derived from CRU? Eventually, please specify. In case you have calculated the correlation based on the different regions results are not appropriate, since for each dataset the regions are different.

- p7, l230: "between the first millennium ...": you actually do not consider the entire millennium, since your simulation starts only in 500 BCE, right? please reformulate

- p7, l234: why non-rotated EOFs in this case?

- p7, l234: again, specify that you conduct the analyses separately for the different seasons. Also, specify the variable that you use as input for the EOF analysis.

- p7, l238: "The index k represents the grid-point index covering the geographical domain": This is not clear. Do you mean that k indicates a given grid box of the domain of study? please reformulate

- p7, l240: "as the noise": is the noise? reformulate

- p7, l241: please justify why using only 3 non-rotated PCs in this case. Depending on the selected variable and season, it seems that the first 3 EOFs do not explain the same amount of variance. Wouldn't it be better to select the number of EOFs depending on a fixed threshold of total explained variance such as the 80%?

- section 3.1: Can you provide more details on how you calculate the spatial correlation between EOFs of each dataset for the different regions? Are you considering the same points in each region?

- p8, l254: Again, can you explain what you mean by paired EOFs? what is the CRU paired with in the first column?

- p8, l256: "For example, GPCC winter region 5 corresponds to GPCC REOF11 as the counterpart to CRU REOF5.": not clear, please reformulate

- p8, l261-262: Reference needed

- p8, l263: the strength of what?

- Fig. 2: What is the darker blue area in the different plots? In each panel it seems that you have more colours than in the legend. Why is that if you only select 6 regions in each case?

- p9, l268-274: are you comparing mean values calculated over different regions in each of the two datasets? if yes, I do not think this is appropriate. For such comparison you should calculate averages in each dataset over corresponding areas and then calculate the differences.

- p9, l273: "..higher than 0.7": It is not clear between what you calculate the correlation. Also, where can I see the values of the correlation and standard deviation in each case? It would be nice to see these values.

- caption of Fig3: what we see in the legend? please add to the caption

- p10, l279: can you better specify, possibly with the help of a figure, how do these precipitation regimes differ?

- p10, l282-284: But this should then be an issue of the GCM, if the issue is related to the large-scale circulation. Could you check if this is the case?

- Table2: why you select again 6 regions?

- p10, l293-294: "Precipitation in NR is mainly concentrated...": I cannot see this from Fig. 4. Please add some climatological characterisation of the different regions in terms of precipitation if you aim to discuss their features.

- p10, l295-297: In the method section it is not clear which model configuration was the base for your simulations. In case you selected the model setup suggested for Europe by the CLM community, you should make this clear in the text. Eventually, you should provide some reason why you did not tune the model properly for the study area, since you mention this as a possible cause of model bias

- p10, l300: also connected to previous questions: if the explained variance is 50% in this case, why not selecting more REOFs?

- Fig.4: why not considering all regions of the domain?

- p11, l308-311: Again, which regions are you considering in the taylor diagrams? the regions of CRU? in this case, please specify it. If you are instead considering the different regions of each dataset, this comparison would not be appropriate for the same reasons given before.

- p11, l307: "it is visible": where can I see this? table 2? please specify

- section 3.2: Why not conducting the analysis over the entire period? also, it would be more appropriate to compare statistics calculated over period of the same length (here 51 against 39 years), in particular when comparing interannual variability

- p13, l340-342: It would be interesting to see a plot of the volcanic forcing over the entire simulation period to better understand the relevance of the ERP with respect to other periods in terms of forcing p14, l348-354: Are these differences particularly different than differences arising from the interannual variability of the entire simulation? you could also have a map of the bias with a dot where the differences are above the standard deviation of the entire simulation for the considered variable and season

- Fig 7: connected to the comment from before: Could you add a map of the mean differences obtained for each variable and season.

- p14, l360-362: Again, you should do these analyses considering periods with the same length

- p14, l372: Reformulate:the NR region is highlighted in blue in Fig. 8. Also, specify to which regions correspond the red and green color boxes

- p15, l393-395: what about summer precipitation and winter temperatures? why not considering them? please specify

- p15, l396: EOFs applied to which variable now? seasonal anomalies of SLP?

- p15, l398: "..the three leading EOFs..": specify of winter precipitation

- section 3.3: as already suggested, EOF analysis of SLP and its projection onto precipitation and temperature should be calculated over the entire period of time. In this way you could assess how relevant changes are for the given periods compared to others, as well as possible trends over the entire period of time.

- p15, l401: which peninsulas? specify

- p15, l401: "...indicating the influence of the westerly circulation and the land-sea interaction": How? can you better specify how this indicates the influence of westerly circulation and land-sea interactions?

- p15, l403: "...likely connected with the direct impact of the large-scale circulation": again, could you better explain how is this plausible?

- p16, l418: did you check cyclones in your data or you have some reference for this statement?

- p18, l453: as for the precipitation, you could extend a bit on the dynamical drivers of the obtained temperature patterns

- p18, 448-458: It is not clear what is causing the differences in dynamics between two periods. Can you make some guesses? Again, here it would be very important to consider the variability of precipitation and temperature for each point over all the simulation period. Or maybe in terms of the EOF, to check how the PCs evolve over the entire simulation period.

- p19, l465: "...we presented the first fully forced CCLM adapted for paleoclimatic applications ...": You actually did not really present it, since you miss lot of crucial information about the model configuration in the methods section

- Conclusion section: you are still missing a comparison against proxies

- p19, l473: "...limitations in representing convective processes linked to the ITCZ.": You actually did not demonstrate this. Based on which ground can you affirm such statement? Eventually, you have all the tools and data for demonstrating this point. For example, you could check how is the ICTZ simulated in your model chain? In case the workload for conducting this analysis would be too much, I would try to be much more careful in explaining the possible reasons of model biases. For example, by extending the description of the paper of Adams et al. 2016, making it clear that this does not discuss an RCM, but the results of GCMs. Again, it would be interesting to see what happens in terms of the ITCZ in your driving GCM MPI-ESM.

- p19, l476477: The same applies to the conclusions of lines 476-477: you have all the data and tools to quickly check whether the drying bias is associated to the representation of clouds in the model.

- p20, l492: "... and is subject to various circulation mechanisms... " Reformulate. In particular, specify which ones are these mechanisms.

- Taylor diagrams in the appendix: The captions of figures in the supplements need some reformulation. Also in the Taylor diagrams it is not clear for which regions you are conducting the analyses. The CRU regions? please specify

---

## Author Comment (AC1)

Review of "The climate of the Eastern 1 Mediterranean and the Nile River basin 2000 years ago using the fully forced COSMO-CLM simulation" by Zhang et al.

This study is a first attempt to mobilize long transient regional climate simulation to understand past climate changes. The study focuses on changes in precipitation and mean 2 m temperature within the Eastern Mediterranean (EM) and Nile River (NR) basin regions from 500 BCE to 1850 CE. The authors evaluate the RCM's capability to simulate homogeneous regions during winter (DJF) and summer (JJA) under current climate conditions (1980-2018), then compare precipitation and 2 m temperature between the Roman period (ERP: 400-362 BCE) and the pre-industrial period (PI: 1800-1850 CE), as well as changes in large scale teleconnection patterns.

I am convinced that dynamical downscaling is a valuable tool to refine coarse GCM outputs to improve our understanding of past climate changes, especially in regions with diverse topography. While the scientific significance is excellent, the methodology used to identify homogeneous regions is not straightforward and the presentation quality deserves to be improved by selecting a subset of results and further highlighting the main take-home messages. Below are a couple of major and minor comments that require consideration for the paper's acceptance for publication in Climate of the Past.

Dear reviewer, thank you very much for the comments and suggestions. We have revised the manuscript accordingly, and please see the following responses for details. In the evaluation part, we have changed the comparison to the datasets ERA-Interim, CRU and CCLM. ERA-Interim is the driving data of the CCLM because the MPI-ESM – LR simulation is only available to the year 2000.

**Major Comments:**

- While the study focuses on winter and summer, the most contrasting seasons, the primary changes between ERP and PI predominantly occur in spring and late summer-fall, especially concerning the mean state (Fig. 7). Including variability (standard deviation for each month for each period and inter-period difference) would provide a more comprehensive understanding of the changes. The authors are encouraged to consider analyzing or discussing the main modes of variability and associated large-scale teleconnections for these seasons as well.

For showing the variability for each month, each period, and each region, as well as comparing the CCLM output to the forcing MPI-ESM, the Box plots are shown on top of the mean annual cycles in Figure R1. Additionally, the associated large-scale teleconnections for four seasons are calculated and plotted and shown in Figure R2 to R5. As illustrated in Figure R1, showing the annual cycle of precipitation and temperature across regions EM, NR, and EMNR, a distinct contrast between the Early Roman Period (ERP) and the pre-industrial (PI) periods is noticeable, particularly from August to January in the case of precipitation. Interestingly, the spatial mean of temperature exhibits minimal differences between the two periods, except during the summer months (July, August, and September).

[Figure]

Figure R1. Precipitation and temperature annual cycle of the PI and ERP periods for EM, NR and EMNR. Monthly mean total precipitation and monthly mean temperature are presented on the left y-axis. Differences between ERP and PI are shown on the right y-axis.

[Figure]

Figure R2. Patterns of the first three non-rotated EOFs of four seasons (DJF, JJA, MAM, SON) precipitation for PI (1800-1850 CE) and ERP (400-362 BCE).

[Figure]

Figure R3. Regression maps between the PCs of the first three non-rotated EOFs for precipitation (four seasons: DJF, JJA, MAM, SON) and SLP from the MPI-ESM simulation for the PI and ERP periods. Only the grid box with statistical significance at the 95 % confidence level is plotted.

[Figure]

Figure R4. Patterns of the first three non-rotated EOFs of four seasons (DJF, JJA, MAM, SON) temperature for PI (1800-1850 CE) and ERP (400-362 BCE).

[Figure]

Figure R5. Regression maps between the PCs of the first three non-rotated EOFs for temperature (four seasons: DJF, JJA, MAM, SON) and SLP from the MPI-ESM simulation for the PI and ERP periods. Only the grid box with statistical significance at the 95 % confidence level is plotted.

- The paper lacks a discussion on the added value of dynamical downscaling compared to the forcing GCM. A comparison of the two solutions in terms of spatial and temporal characteristics of homogeneous regions (e.g., Figs. 2-3) and annual cycles (Fig. 7) would help illustrate the usefulness of long transient RCM simulations for paleoclimate.

For providing the added value of the dynamical downscaling compared to the forcing data, we have changed the evaluation part by comparing CRU (observational data), Era-Interim (forcing global data) and CCLM (regional climate modelling output) to further address the added value of dynamical downscaling of global climate modelling date see Figure R6 to R9 for both EM and NR region. In addition to this we have also compared the annual cycle of the three data set for both precipitation and temperature, see Figure R10.

[Figure]

Figure R6. Regionalization of precipitation over the EM in winter (DJF, upper panel) and summer (JJA, lower panel) for CRU (a, d), ERA (b, e) and CCLM (c, f). The total explained variance of the corresponding REOF is shown in the legend.

[Figure]

Figure R7. Seasonal mean total precipitation differences of EM regions with respect to the 1980-2018 mean total CRU winter (DJF) and summer (JJA) precipitation.

[Figure]

Figure R8. Regionalization of precipitation over the NR in winter (DJF, upper panel) and summer (JJA, lower panel) for CRU (a, d), ERA (b, e) and CCLM (c, f). The total explained variance of the corresponding REOF is shown in the legend.

[Figure]

Figure R9. Seasonal mean total precipitation differences of NR regions with respect to the 1980-2018 mean total CRU winter (DJF) and summer (JJA) precipitation.

- Changes in annual cycle and teleconnection patterns presented in Section 3.2 could gain in robustness by evaluating the RCM's capability to simulate these characteristics under current climate conditions. Fig. 7 should be duplicated for the 1980-2018 period (Obs + RCM + GCM) and included at the beginning of the evaluation section.

Figure 7 has been duplicated for the 1980-2018 period as well, see Figure R10, where the CRU data is the observational data, ERA is the global forcing data of regional climate model CCLM.

[Figure]

Figure R10. Precipitation and temperature annual cycle of the present period for EM, NR and EMNR of CRU, ERA and CCLM.

- Teleconnections should also be evaluated for the 1980-2018 period in the Supplementary Material and briefly discussed in the main text.

We also present the results of the teleconnections of the 1980-2018 period in Figure R11- R14. This calculation was done when we were preparing the manuscript, it will be provided in the supplementary if necessary. As discussed in the manuscript, the relationship between the sea level pressure and summer precipitation over the EM region does not show a clear pattern as well. Furthermore, only the first EOF of the winter precipitation over the EM region shows a relation with the low pressure center over the EM region and strong positive sea level pressure over the northern Atlantic ocean.

[Figure]

Figure R11. Patterns of the first three non-rotated EOFs of DJF and JJA precipitation for 1980-2018 CE.

[Figure]

Figure R12. Regression maps between the PCs of the first three non-rotated EOFs for precipitation (seasons: DJF, JJA) and SLP from the ERA-Interim of 1980-2018 period. Only the grid box with statistical significance at the 95 % confidence level is plotted.

- Regarding the EOF approach, there are a couple of questions:

1. The rationale for retaining only grid points with positive values for identifying homogeneous region needs clarification, considering that EOF loadings can include strong negative and positive values simultaneously e.g. in the case of a dipole pattern. The possibility of one specific grid point being classified in more than one region should be addressed, along with the sensitivity to the percentile threshold.

   The identification of the homogeneous region is done based on the EOF pairs (Table 1 and 2 in the manuscript) which we have calculated with reference to CRU data. Further, the homogeneous regions between the three data set were identified by setting a threshold. Where the value of one specific grid is equal or greater than the threshold value is part of the subregion. We have chosen the threshold value as $80^{th}$ ($75^{th}$) percentile of the precipitation (temperature) loading. As the threshold is set as the percentile of the precipitation (temperature), thus strong negative and positive values when considering the EOF loadings with a dipole pattern are all considered as well.

   Here we are presenting the results of using different percentiles (EM region as an example). In Figure R13, we can see that by setting different thresholds, the subregion experiences more overlap when the threshold is too low and covers only small parts of the domain when the threshold is too high. In addition, in Table R1 and R2, we are showing the standard deviation and the percentage of the grid points which have been retained as a subregion. This justifies the selected thresholds in our manuscript.

[Figure]

Figure R13. Subregions when using different thresholds for the EM region.

Table R1. Standard deviation of the subregions when using different thresholds.

|         | Region-1 | Region-2 | Region-3 | Region-4 | Region-5 | Region-6 |
|---------|----------|----------|----------|----------|----------|----------|
| CRU-60  | 11.27    | 8.30     | 11.16    | 9.53     | 6.88     | 8.73     |
| CRU-80  | 10.06    | 6.68     | 10.58    | 7.08     | 4.98     | 8.30     |
| CRU-90  | 9.09     | 5.18     | 9.21     | 4.92     | 2.95     | 6.86     |
| ERA-60  | 10.30    | 6.80     | 10.29    | 9.31     | 4.58     | 8.79     |
| ERA-80  | 8,24     | 4.74     | 9.82     | 7.05     | 3.69     | 10.33    |
| ERA-90  | 7.37     | 2.97     | 8.47     | 5.05     | 1.94     | 9.40     |
| CCLM-60 | 6.14     | 8.47     | 12.10    | 6.07     | 2.66     | 10.29    |
| CCLM-80 | 4.74     | 7.13     | 12.69    | 6.56     | 1.97     | 12.88    |
| CCLM-90 | 3.81     | 5.60     | 12.23    | 5.54     | 0.32     | 14.03    |

Table R2. Covered grid box percentage of the whole region of the subregions when using different thresholds.

|         | Region-1 | Region-2 | Region-3 | Region-4 | Region-5 | Region-6 | sum   |
|---------|----------|----------|----------|----------|----------|----------|-------|
| CRU-60  | 0.32     | 0.33     | 0.20     | 0.18     | 0.16     | 0.13     | 1.32  |
| CRU-80  | 0.19     | 0.21     | 0.08     | 0.10     | 0.07     | 0.03     | 0.68  |
| CRU-90  | 0.09     | 0.09     | 0.05     | 0.06     | 0.02     | 0.02     | 0.33  |
| ERA-60  | 0.34     | 0.30     | 0.23     | 0.19     | 0.12     | 0.14     | 1.32  |
| ERA-80  | 0.19     | 0.17     | 0.10     | 0.12     | 0.04     | 0.04     | 0.66  |
| ERA-90  | 0.12     | 0.07     | 0.05     | 0.07     | 0.009    | 0.02     | 0.331 |
| CCLM-60 | 0.29     | 0.35     | 0.22     | 0.13     | 0.17     | 0.18     | 1.34  |
| CCLM-80 | 0.17     | 0.23     | 0.11     | 0.05     | 0.04     | 0.07     | 0.67  |
| CCLM-90 | 0.06     | 0.15     | 0.07     | 0.02     | 0.001    | 0.03     | 0.331 |

2. Considering the lengthy model evaluation section, it might be advantageous to perform one EOF analysis (or a more straightforward method: see my next comment) for the entire region (all grid points within the EM-NR region) and the entire annual cycle (monthly anomalies) simultaneously. Alternatively, some results could be moved to the Supplementary Material to shorten the section and to let room for a discussion on RCM added values compared to its forcing GCM.

During the preparation of this manuscript, we have tried first to use the entire region to perform the EOF analysis. Due to very different precipitation characteristics of the EM and NR region, the EOF analysis cannot identify the variabilities when using the entire region. Thus, we have decided to separate the region.

3. The EOF approach is not straightforward to identify homogeneous regions in terms of temporal variability. As illustrated in Figures 3 and 5, it becomes apparent that certain regions exhibit highly similar temporal variability. I believe the paper could gain valuable insights by either transitioning to or, at least, discussing clustering analyses (e.g., k-means, hierarchical clustering) since these techniques have proved to efficiently identify regions with shared temporal variability.

To better demonstrate the reason why we are using EOF analysis, we will add a paragraph discussing about those different methods. Compared to machine learning based clustering, regionalization based on EOF remains the physical meaning and the principal components retained from the EOF can further help to demonstrate the temporal variation and link to large scale circulation in climate research.

4. The rationale behind performing rotated EOFs for the present-day period while classical EOFs for the PI and ERP periods should be clarified.

As we mentioned in the manuscript. The classic patterns, EOFs, are orthogonal, hence, the eigenvectors are uncorrelated. For some applications, this is a useful characteristic (i.e., setting up multiple regression models with predictors that are not collinear). In meteorological applications, however, the orthogonality constraint may be disadvantageous, because most processes in the real world are not orthogonal (Storch and Zwiers, 1984). We thus applied the VARIMAX rotation to obtain rotated EOFs (REOFs) that are physically more consistent than the non-rotated patterns. REOFs are thus used for the validation of the model set-up in the present period (1980 - 2018 CE). While non-rotated EOFs are implemented for the comparison of the mean climate conditions in PI and ERP times, ensuring the preservation of meteorological characteristics in temperature and precipitation. The application of non-rotated EOFs guarantees the accurate interpretation of the regression involving the non-rotated principal components of precipitation and temperature with respect to large-scale circulation patterns.

5. Important details about the RCM simulation are missing in Section 2.1, such as the horizontal resolution and more information on changes in forcings, particularly land cover changes over the 2500-year period. Additionally, the high correlation of simulated precipitation's temporal variability with observations (Figs. 3 and 5) requires clarification, given the RCM simulation is forced with outputs from an ocean-atmosphere coupled GCM.

We are currently preparing a paper which is mainly discussing the set up and the changes in the different external forcing in detail (Hartmann, et al. 2024). We can share the manuscript confidentially to further illustrate the above-mentioned information. The

external forcings are based on the recommendation for the PMIP4 past1000 contribution to CMIP6 (Jungclaus et al., 2017). We will add some more information in Section 2.1. Here is some more detailed information:

The orbital forcing is represented by the eccentricity, the obliquity and the longitude of perihelion.

The total solar irradiance represents the solar forcing.

The volcanic by the aerosol optical depth.

The changes of greenhouse gas concentrations are given in equivalent CO2 concentrations which consists of CO2, CH4 and N2O.

The land use changes are introduced as LAI, maximum and minimum plant coverage derived from the MPI-ESM transient simulation.

**Minor comments**

All the minor comments will be addressed as suggested.

---

## Author Comment (AC2)

Review Zhang et al., "The climate of the Eastern Mediterranean and the Nile River basin 2000 years ago using the fully forced COSMO-CLM simulation".

General comments

The paper by Zhang et al. presents results from a transient climate simulation covering 500 BCE to 2018 CE. This simulation has been performed with the paleoclimate version of the COSMO-CLM regional climate model (RCM), which was forced with a global simulation carried out with the MPI-ESM-LR model. Such a transient simulation with an RCM is novel and provides a spatially detailed view of the climate. In this paper, the RCM is set up for a domain spanning the Nile River basin and the eastern part of the Mediterranean. The results section consists of two parts. In the first part, the model's performance is evaluated by comparing the simulated climate for the period 1980-2018 with observations and reanalysis data for the same time frame. This comparison indicates that the model is doing a reasonable job, except for the precipitation in the Nile River domain. The second part provides a comparison of the results for two periods (the Early Roman Period (ERP, 400-362 BCE) and the preindustrial period (PI, 1800-1850 CE)), and presents also an analysis of the link between temperature and precipitation on the one hand, and the atmospheric circulation on the other hand.

Although the performed simulation is innovative and represents a technological improvement that could play an important role in studies of past climates, there are several issues with this paper that need to be resolved before it can be published. These issues are discussed in detail below, and relate to the purpose of the paper, the set-up of the simulation and the analysis of the results.

Thank you very much for taking the time to review out manuscript. We will modify the manuscript accordingly, please see the following responses to the different comments. The primary aim of the paper is to provide a comprehensive characterization of the ERP climate. To achieve this, we, in the first part of the manuscript, focusing on the evaluation of the CCLM output. Additionally, we acknowledge the necessity of refining the employed methodology to offer a more accurate description of the applied analysis. It's essential to clarify that the primary focus of the paper is to compare the BCE climate to the pre-industrial climate and further characterize the climate over EM and NR during the BCE period.

**Main comments**

Purpose of the paper. It is not clear to me what the purpose of this paper really is. One goal is to provide an evaluation of the performance of the model for the present-day climate, but this evaluation is not presented as being the main aim. The results for the ERP and PI periods are compared, and it may be expected that a characterization of the ERP climate is the main objective of this paper, as the PI period is usually taken as the reference period in paleoclimate modelling. However, the introduction section of this paper does not even mention the Early Roman Period once, and thus does not explain at all why there is a focus on the ERP. My advice is therefore to streamline the introduction section and make a case why it is important to obtain a more detailed climatological understanding of the ERP. This should include an explanation of why the 400-362 BCE period is chosen. Why not another period, and why these specific 38 years? To my knowledge, there is no established definition of the Early Roman Period that falls between 400 and 362 BCE, so this requires a thorough discussion.

Thank you very much for the suggestions regarding on how to more concisely present the purpose of the manuscript. The purpose of this paper includes two parts, part one: an evaluation of the performance of the model for the present-day climate; part two: compare the ERP and PI periods, and a characterization of the ERP climate. We agree with your

opinion of adding a paragraph in the "Introduction" section to thoroughly discuss the reason of why we are choosing the ERP period specifically. For choosing these two periods, we have analyzed the time series from 500 BCE to 1850 CE of the MPI-ESM output and the forcing data. In the study of (McCormick et al., 2012a), one have tried to explore the relationship between climate change during the Roman Empire period. Nevertheless, this study only goes to 100 BCE while the entire roman empire starts back to around 500 BCE. Thus, together with comparing the long time series of precipitation and temperature from the MPI-ESM output, we choose this period: 400-362 BCE as this period goes early back to the time when the roman empire is rising from a kingdom. In addition to this, the volcanic activities (Toohey and Sigl, 2017) are also taken into consideration for choosing these two periods.

Given the substantial computational and time resources required for this process, the Regional Climate Model (RCM) paleo transient simulation results spanning the entire 2500-year period is not available now, therefore comparing the entire period output with proxy records is in our future study scope.

Set-up of the simulation - CCLM. The results are obtained with the newly developed paleoclimate version of the CCLM model. However, it is not explained what has been modified compared to the "normal" version. More details on the model should thus be provided. What is meant by the phrase "by implementing CCLM in model version 5.0 with CLM version 16"? this phrase suggests that CCLM is a different model from CLM, but CCLM just stands for COSMO-CLM, doesn't it? And what is actually the spatial resolution of the version of CCLM applied here?

We have implemented the external forcings (solar, orbital, volcanic, GHG and land use change) to the normal version of CCLM. The reason for this is that in paleoclimate applications of CCLM, those forcings have never been fully implemented while they are in the driving models. For example in some papers, when it is concerning the climate of LGM (Ludwig et al., 2016), one has implemented the orbital forcing specifically, and in some other research regarding the GHG, one have implemented the GHG forcing into CCLM. But until now, there is not such a simulation using CCLM with all the forcing implemented. Thus, we have decided to run a fully forced CCLM simulation with all the external forcings.

Regarding the confusion of the phase: "by implementing CCLM in model version 5.0 with CLM version 16", here we intend to say by implementing are the external forcing into the CCLM (COSMO model version 5.0 with CLM version 16). We are going to correct that also in the manuscript.

For choosing the spatial resolution for this simulation (0.44 °), we did test simulations with 0.11° and 0.44° horizontal resolution. By comparing the results of simulation output for these two different spatial resolutions, we did not find much added value of higher resolution at 0.11 in the regard of presenting the regional precipitation and temperature. But instead, by improving the horizontal resolution to 0.11, the computation time and storage resources consumed is much higher (16 times) than 0.44 °. Thus, we have chosen 0.44 ° for this specific simulation.

Set-up of the simulation – applied forcings. Several forcings are applied in the simulation, but it is not clear what the main differences between these forcings are. The authors mention orbital, solar, GHG, volcanic and land-use changes. I suggest explaining what the main differences in radiative forcings are between the main periods of interest in this paper: present period, PI and ERP. How do each of these radiative forcings change between the three periods? What forcings are most important, and how large are the differences when expressed in Wm-2? This information is important for interpreting the results and should be included.

We are currently preparing a paper which mainly discussing the set up and the changes in the different external forcing (Hartmann, et al. 2024). Due to the different research focus, regarding the detailed information of the forcing are presented in Hartmann, et al.  2024. We can share the manuscript confidentially to further illustrate the above-mentioned information. The external forcings are based on the recommendation for the PMIP4 past1000 contribution to CMIP6 (Jungclaus et al., 2017). Here is some more detailed information:

The orbital forcing is represented by the eccentricity, the obliquity and the longitude of perihelion. Those values constantly change with time. The effect of the orbital forcing is further explained in the next response.

The total solar irradiance represents the solar forcing. The ERP is in a solar minimum phase where TSI values maximum as high as the lowest values of PI. Today's values are similar to PI.

The volcanic by the aerosol optical depth. Both periods cover silent periods and larger volcanic eruptions.

The changes of greenhouse gas concentrations are given in equivalent CO2 concentrations which consists of CO2, CH4 and N2O. The PI has higher values as the ERP. For the present-day simulation we use only CO2 concentrations at the level of 1950, which is lower than both equivalent CO2 concentrations.

The land use changes are introduced as LAI, maximum and minimum plant coverage derived from the MPI-ESM transient simulation.

Analysis of the results. The differences between the ERP and PI climates should be discussed in terms of the differences in radiative forcings between the two periods (see point 3 above). In addition, it should become clear how the results for the ERP compare to proxy-based climate reconstructions for the same period. On line 497 it is noted that "comparing modeled data with proxy records is essential for a comprehensive understanding of climate variability", but surprisingly the authors do not make such a comparison for the results presented in this paper. This is clearly a missed opportunity, and in my opinion a model-proxy comparison should be included. Moreover, the model results should also be discussed more extensively relative to previous modelling studies.

We will include the information of radiative forcing of the two periods as well as in the discussion. Nevertheless, as we also plan to submit a paper dedicated to explaining the setup and external forcings, we will only bring up the difference of external forcing such as radiative and volcanic over the two periods. For details regarding the forcings please refer to the response to the "set up of the simulation".

For the further information about the radiative forcings between the two studied period, we will include a new plot here which shows the differences of the insolation of the two periods (see Figure R1). For the volcanic forcing, we have used the reconstruction from Toohey and Sigl, 2017.

[Figure]

Figure R1. The insolation difference between the centuries of ERP and PI caused by different orbital forcing values.

As comparing to the proxy records. We completely agree that it will be largely important to the whole study, as we are currently analyzing the data. Nevertheless, given the substantial computational and time resources required for this process, the Regional Climate Model (RCM) paleo transient simulation results spanning the entire 2500-year period is not available now, therefore comparing the entire period output with proxy records is in our future study scope.

Additionally, this paper, has a different focus in evaluating the RCM simulation results and characterized the ERP climate. We will include in the discussion how this study can benefit the future study such as comparing the simulation output to proxy records. Nevertheless, we will modify the introduction and give a more consistent presentation of the

overall subject of the paper. We are working specifically to and preparing a manuscript which is dedicated to compare our simulation output with the proxy records.

Discussion. The present paper presents the first transient RCM simulation of the past 2500 years, so one question to answer is if there is a clear advantage of making the effort of running an RCM in transient mode. If the authors would run three snapshot RCM experiments for present-day, PI and ERP, to what extent would the results be different? In other words, what have we learned about the ERP climate from the transient simulation that could not be obtained from a snapshot experiment with mean forcings for the ERP?

The added value of running the RCM in a transient mode will be discussed in detail another manuscript we are preparing currently, as mentioned in the response to comment: set up of the simulation. In that paper, the impacts of running a transient RCM with all the forcings are presented. If necessary, we can share the manuscript with the reviewer confidentially in order to resolve the confusion mentioned in the comment.

**Other comments**

All the other comments will be addressed as suggested. Especially the plots with significance dot have been modified by only plotting the significant grid box.

---

## Author Comment (AC3)

**Review of "The climate of the Eastern Mediterranean and the Nile River basin 2000 years ago using the fully forced COSMO-CLM simulation" by Zhang et al. 2023**

**General Comments**

In this manuscript the authors present the results of the first fully-forced, high resolution simulation of the climate of the Eastern Mediterranean and Nile river regions over the last 2500 years using the COSMO-CLM regional climate model. The work is mainly divided in two parts. The first one, where the authors evaluate the model with several modifications necessary for the simulation of past periods, for the present-day. And the second one where they assess climate changes between two past periods of time, namely the Early Roman Period (ERP) and the Pre-Industrial (PI) period. Here they assess differences in seasonal values of precipitation and temperature, as well as their connection to changes in the atmospheric circulation.

The paper presents some interesting results and its contents fit well within the scopes of Climate of the Past. Nonetheless, I do believe that the manuscript suffers from a series of major issues that need to be properly addressed before it could be considered for publication for the journal.

First of all, the objectives of the paper are in my opinion not very well defined. The employed methods are not always clearly described, both concerning the description of the experimental design of the presented simulations as well as for the statistical methods employed in the analysis of their results. This makes the understanding of the different analysis not always straightforward. Additionally, the presented analysis can sensibly be extended, making a full use of the transient simulation and of the driving GCM data, to understand discriminated model biases, as well as possible differences in the climate of different time frames of the simulation period. The beauty of your simulations is that you have so much data from which we could really learn a lot about past climate changes of the investigated area and their drivers. I would try to make a full use of them. Finally, it would be interesting to know how the model performs in past times, performing a comparison against proxy data for the study period. Please, find below more detailed comments on which I based my judgement.

Thank you for your insightful comments and suggestions on the manuscript. We have thoroughly reviewed each of your points and are committed to addressing them effectively.

In response to your feedback regarding the objectives of the paper, we recognize the need for modification. The primary aim of the paper is to provide a comprehensive characterization of the ERP climate. To achieve this, we, in the first part of the manuscript, focusing on the evaluation of the CCLM output. Additionally, we acknowledge the necessity of refining the employed methodology to offer a more accurate description of the applied analysis. It's essential to clarify that the primary focus of the paper is to compare the BCE climate to the pre-industrial climate and further characterize the climate over EM and NR during the BCE period. We want to highlight that covering the entire period of the transient simulation is beyond the scope of this specific study. Furthermore, we share your interest in comparing proxy data to our CCLM transient simulation output. We are currently in the process of preparing a dedicated manuscript specifically addressing the aspects related to proxy data. For a more detailed information, please see the following responses to the major comments.

**Major comments**

- 1. p3, l85-87: you mention that for this area a dense network of natural archives is available covering the last 2000 years. These data should be acknowledged and used for the comparison against your model results.

  We agree with this opinion, the data is currently under processing and being compared with the proxy record. Nevertheless, the purpose of this paper is composed by two parts, one is to provide an evaluation of the performance of the model for the present-day climate, and the other is to compare the ERP and PI periods, further provided a characterization of the ERP climate. We will modify the purpose of the manuscript and revised the introduction by explaining why there is a focus on the ERP period and why it is important to obtain a more detailed climatological understanding of the ERP. And for further comparing the simulated output with proxy is not within the study scope of this manuscript but another direction that we are currently working on.

- 2. p4., l133: you need to present a summary of the model setup, particularly concerning information on how you implemented changes in the model to take into account changes in the forcing.

  We are currently preparing a paper which is mainly discussing the set up and the changes in the different external forcing in detail (Hartmann, et al. 2024). We can share the manuscript confidentially to further illustrate the above-mentioned information. The external forcings are based on the recommendation for the PMIP4 past1000 contribution to CMIP6 (Jungclaus et al., 2017). We will add some more information in Section 2.1. Here is some more detailed information:

  - The orbital forcing is represented by the eccentricity, the obliquity and the longitude of perihelion.

  - The total solar irradiance represents the solar forcing.

  - The volcanic by the aerosol optical depth.

  - The changes of greenhouse gas concentrations are given in equivalent $CO_2$ concentrations which consists of $CO_2$, $CH_4$ and $N_2O$.

  - The land use changes are introduced as LAI, maximum and minimum plant coverage derived from the MPI-ESM transient simulation.

- 3. section 2.3.1: since you have data, wouldn't it be better to conduct EOF analyses of seasonal anomalies over the entire simulation time period and detect possible trends? In this way you could also compare changes across different periods. I think that this analysis, considering the fact that the study presents for the first time the results of a transient simulation for the area at high resolution, would be quite interesting. Also, are the presented results sensitive to the relatively short length of the two time periods considered?

  We acknowledge the suggestion to conduct EOF analyses across the entire period for seasonal anomalies. However, in the specific context of characterizing the climate of the ERP periods, we find it necessary to perform EOF analyses over the selected periods. This is essential for achieving the main objective of comparing the ERP climate to the pre-industrial (PI) period.

  We have analyzed the trend of the entire period, based on General Circulation Model (GCM) output, which helps us to identify the BCE period for further investigation. Given the established connection between climate conditions and societal events (McCormick et al., 2012) and notably, this specifically

focuses on climate change and the Roman Empire back to 100 BCE, leaving a gap concerning the earlier Roman Kingdom period, which dates back to around 500 BCE.

To address this gap and explore associations between climate change and the entire Roman period, we have in this study chosen the period 400-362 BCE. This period predates the established Roman Empire, capturing the rise from a kingdom. By examining the characteristics BCE period temperature and precipitation in this study, we can provide knowledge of the climate back to 2000 years ago in a more detailed spatial and temporal resolution.

- 4. section 2.3.2: the method you use for the clustering of the different regions according to seasonal values of precipitation and temperature is not entirely clear. This part needs to be revised and possibly extended with additional details.

To address this comment, we have made substantial revisions to the method section. Additionally, we have added discussions on different cluster methods in both the introduction and discussion sections to provide a comprehensive overview of our approach.

Also, we will provide additional details that support the specific procedures employed in this study. For a thorough understanding of these enhancements, please refer to our response in comment 6. We believe these adjustments significantly contribute to the clarity and robustness of our methodology, ensuring a more detailed presentation of our research approach.

- 5. section 2.3.2: Additionally, there are many choices that seem arbitrary in your method and that need further testing: for example, l219-220, why choosing only 6 EOFs for CRU and all for the other datasets?

We chose the CRU dataset as our reference data for assessing the simulation output in the first part of the manuscript. We chose the first six principal components because with these first six components, we cover more than 75% of the dataset's variability. Given that the purpose of principal component analysis captures the major data patterns, using beyond the sixth component will not give valuable information for analysis. This choice guarantees that we are keeping the most important information of the dataset, helping us to better identify the most important precipitation and temperature in the subsequent analysis.

- 6. section 2.3.2: Another important point: do the different regions you derived from the different datasets contain different number of grid-boxes? This is a point that needs particular attention, in particular for the conclusions you draw from Fig. 3 and Fig. 5. When you quantify the match between datasets across regions, as performed in Fig. 5 and Fig. 3, you need to consider overlapping regions with the same number of points.

The overlapped grid boxes of the different subregion should be taking into consideration. Homogeneous region identification is based on the calculated EOF pairs with reference to the CRU dataset (refer to Tables 1 and 2 in the manuscript). Subsequently, we establish homogeneous regions across the three datasets by employing a threshold criterion. Specifically, grid points with values equal to or exceeding the chosen threshold are designated as part of the subregion. For this study, we set the threshold at the 80th percentile for precipitation and the 75th percentile for temperature, determined from the respective loading distributions. This percentile-based thresholding considers both strong negative and positive values, accommodating dipole patterns in the EOF loadings.

Therefore, for better illustrating the choice of the threshold in order to have less grid boxes being retained in more than one subregion, we will therefore present results using different percentiles as an example, focusing on the EM region. Figure R1 demonstrates that different thresholds influence subregion overlap, with lower thresholds causing more overlap and higher thresholds resulting in less coverage. Additionally, Table R1 and R2 provide insights into the standard deviation and the percentage of retained grid points

for each subregion. These details contribute to a comprehensive understanding of the subregion identification process.

[Figure]

Figure R1. Subregions when using different thresholds for the EM region.

Table R1. Standard deviation of the subregions when using different thresholds.

| std | Region-1 | Region-2 | Region-3 | Region-4 | Region-5 | Region-6 |
|---|---|---|---|---|---|---|
| CRU-60 | 11.27 | 8.30 | 11.16 | 9.53 | 6.88 | 8.73 |
| CRU-80 | 10.06 | 6.68 | 10.58 | 7.08 | 4.98 | 8.30 |
| CRU-90 | 9.09 | 5.18 | 9.21 | 4.92 | 2.95 | 6.86 |
| ERA-60 | 10.30 | 6.80 | 10.29 | 9.31 | 4.58 | 8.79 |
| ERA-80 | 8,24 | 4.74 | 9.82 | 7.05 | 3.69 | 10.33 |
| ERA-90 | 7.37 | 2.97 | 8.47 | 5.05 | 1.94 | 9.40 |
| CCLM-60 | 6.14 | 8.47 | 12.10 | 6.07 | 2.66 | 10.29 |
| CCLM-80 | 4.74 | 7.13 | 12.69 | 6.56 | 1.97 | 12.88 |
| CCLM-90 | 3.81 | 5.60 | 12.23 | 5.54 | 0.32 | 14.03 |

Table R2. Covered grid box percentage of the whole region of the subregions when using different thresholds.

| | Region-1 | Region-2 | Region-3 | Region-4 | Region-5 | Region-6 | sum |
|---|---|---|---|---|---|---|---|
| CRU-60 | 0.32 | 0.33 | 0.20 | 0.18 | 0.16 | 0.13 | 1.32 |
| CRU-80 | 0.19 | 0.21 | 0.08 | 0.10 | 0.07 | 0.03 | 0.68 |
| CRU-90 | 0.09 | 0.09 | 0.05 | 0.06 | 0.02 | 0.02 | 0.33 |

| ERA-60 | 0.34 | 0.30 | 0.23 | 0.19 | 0.12 | 0.14 | 1.32 |
|--------|------|------|------|------|------|------|------|
| ERA-80 | 0.19 | 0.17 | 0.10 | 0.12 | 0.04 | 0.04 | 0.66 |
| ERA-90 | 0.12 | 0.07 | 0.05 | 0.07 | 0.009 | 0.02 | 0.33 |
| CCLM-60 | 0.29 | 0.35 | 0.22 | 0.13 | 0.17 | 0.18 | 1.34 |
| CCLM-80 | 0.17 | 0.23 | 0.11 | 0.05 | 0.04 | 0.07 | 0.67 |
| CCLM-90 | 0.06 | 0.15 | 0.07 | 0.02 | 0.001 | 0.03 | 0.33 |

- 7. section 2.3.2: Why for the present-day you use rotated EOFs and for the investigation of past periods you use non-rotated ones?

  As we mentioned in the manuscript. The classic patterns, EOFs, are orthogonal, hence, the eigenvectors are uncorrelated. For some applications, this is a useful characteristic (i.e., setting up multiple regression models with predictors that are not collinear). In meteorological applications, however, the orthogonality constraint may be disadvantageous, because most processes in the real world are not orthogonal (Storch and Zwiers, 1984). We thus applied the VARIMAX rotation to obtain rotated EOFs (REOFs) that are physically more consistent than the non-rotated patterns. REOFs are thus used for the validation of the model set-up in the present period (1980 - 2018 CE), while non-rotated EOFs are implemented for the comparison of the mean climate conditions in PI and ERP times, ensuring the preservation of meteorological characteristics in temperature and precipitation. The application of non-rotated EOFs guarantees the accurate interpretation of the regression involving the non-rotated principal components of precipitation and temperature with respect to large-scale circulation patterns.

- 8. section 2.3.3: since you have the results, why not showing the analysis in temperature, precipitation and mean sea level pressure for the entire simulation period? I think this would give some important and interesting insights on the simulated climate of the given period and area. In any case, whenever you show the differences between the two selected periods you must use the transient results for the entire simulation period to assess whether the obtained differences are comparable to the ones of other periods or if they particularly stand out? in the latter case, you could eventually try to assess why.

  Thank you very much for pointing this out, we agree that it will be very interesting to investigate the temperature, precipitation, and sea level pressure for the entire simulation period. Nevertheless, the purpose of this paper is 1: to assess the transient simulation and 2: describe the characteristics of the ERP climate. Thus, for the EOF analysis, if we select the whole simulation period, we cannot characterize the ERP climate specifically but an overall trend over the last 2500 year. Nevertheless, for choosing the current two interesting period, we have analyzed the trend of the entire period, based on General Circulation Model (GCM) output, for more details regarding why we have chosen this two specific period, please refer to the response of comment 3. Given the substantial computational and time resources required for this ongoing process, the Regional Climate Model (RCM) paleo transient simulation is currently in progress. Consequently, the simulation results spanning the entire 2500-year period will not be available in the near term.

- 9. section 2.3.2 and 2.3.3: In the paper the authors do not acknowledge in any way how the outcomes and conclusions of the manuscript are subject to the series of different arbitrary criteria they applied throughout their analysis. At least some discussion is needed here, to make readers aware that some changes might occur when changing some details of the method.

  We are aware of the different clustering methods like K-means and Hierarchical clustering. To enhance clarity on our choice of employing Empirical Orthogonal Function (EOF) analysis, we plan to include a

paragraph discussing these alternative methods. Unlike machine learning-based clustering, regionalization based on EOF analysis retains a strong physical interpretation. Furthermore, the principal components derived from EOF analysis provide valuable insights into temporal variations and their connections to large-scale circulation patterns. This approach aligns with our goal of not only capturing patterns but also understanding the underlying physical processes driving regionalization.

- 10. Fig3 (same for Fig. 5): Why not comparing first the mean regional climatological values for a given region between the different datasets and then comparing the anomalies of each time series calculated with respect to the corresponding mean value of each dataset. Basically, instead of calculating all the anomalies with respect to the mean value of CRU in each region, it would be more appropriate to remove from the time series of each dataset the corresponding mean for the calculation of the seasonal anomalies. In this way you would have a proper assessment of the differences in the mean in each dataset as well as in their temporal variability.

  We understand that comparing mean regional climatological values first and then calculating anomalies with respect to the corresponding mean value for each dataset is indeed a valid approach. In the first part of the manuscript, we want to validate our simulation results. By calculating seasonal anomalies with reference to CRU data, we can not only obtain a more accurate assessment of differences in both the mean and temporal variability across datasets but also if there is a systematic bias of our simulation results against the observational/reanalysis data sets (such as overestimation or underestimation).

  This methodology ensures a proper evaluation of how each dataset deviates from the observational data (CRU), providing insights into the characteristics and variability within each dataset. In addition to this,

  We compared the absolute monthly total precipitation and mean temperature between the GCM, observational data and CCLM for period 1980-2018 to further demonstrate the mean state of the different data sets, see Figure R2, where the CRU data is the observational data, ERA is the global forcing data of regional climate model CCLM.

[Figure]

- 11. Fig. 7: Why are you now simply comparing spatial means over the entire region? in particular, what is the need for all the previously conducted analyses on sub-regions that you performed in previous sections in this context?

In Figure 7, we present the mean values across EM, NR, and the combined EMNR as a comprehensive comparison with the pre-industrial period. Additionally, we employ Empirical Orthogonal Function (EOF) analysis to explore the connection between major precipitation/temperature patterns and sea level pressure. The rationale behind assessing the CCLM output through clustering via EOF analysis is to scrutinize its efficacy in capturing temporal variance within specific subregions.

Initially, we attempted to perform EOF analysis over the entire region. However, due to distinct precipitation characteristics in EM and NR regions, the analysis struggled to identify variabilities specific to each. Consequently, we choose to separate the region, recognizing the unique patterns within EM and NR. Moreover, considering the different precipitation regime of the two regions, the application of EOF methods might yield different results. By clustering the region based on EOF analysis, we could evaluate the simulation results by looking at that if the model is reasonably simulation the climate of the different variabilities.

12. section 3.3: Alternatively, you could also consider to conduct a canonical correlation analysis between SLP and precipitation and temperature over the entire period of time.

In examining large-scale circulation patterns associated with principal components, there are several methods available, including canonical correlation analysis and linear regression. In our specific approach, we utilized Empirical Orthogonal Function (EOF) analysis to derive the primary principal components of precipitation and temperature data. Given this, we have opted for linear regression in this study to investigate the potential relationship between the major precipitation/temperature EOF pattern and sea level pressure. This choice aligns with our aim to understand the specific associations between these climatic variables and provides a focused exploration of their interconnections through linear regression analysis.

**Minor comments**

All the minor comments will be addressed as suggested. Specially, response to some of the minor comments are described here.

- p1, l17: you do not develop COSMO-CLM. You rather apply a high- resolution climate model modified for its application to paleoclimate studies. Make sure in the text that some studies already applied modified versions of COSMO-CLM to paleoclimate.

  Sentence reformulated as: We modified a high-resolution regional model, COSMO-CLM, for paleoclimate applications, by integrating all external forcings and conducted a transient simulation from 500 BCE to 1850 CE.

- p3, l97-104: I miss here some discussion, also based on previous literature, on why the application of RCMs to the study of the past is relevant.

  While the proxy- based climate reconstructions are location based and spread over several locations within the interested region. A RCM could help to close this gap and provide a better representation of the spatial patterns.

- p4, l119: I guess just some of the Nile flooding match volcanic eruptions and not all? maybe it might be interesting to report some example?

  Researchers looked for the relationship between the Nile flood anomaly from climate modelling output with the volcanic eruption in 1902, 1912, 1963 and 1982. And found out that the Nile floods matched

with the effects of volcanic eruptions, i.e., large-scale volcanic eruptions that cool the Earth's atmosphere and thus disrupt the normal flow of the Nile.

- p4, l120: the increase in energy in the climate system is continuous only for a continuous increase in GHGs. Please modify accordingly.

  Modified: Continuous increase of Greenhouse gases (GHGs) trap in turn the longwave radiation that is emitted by the Earth surface and lead to a continuous increase of energy in the climate system and effects on the global climate (Ramanathan and Feng, 2009)

- p4, l127: "but those forcings are not yet fully implemented in the RCM": please be aware that many other studies with modified forcing were already performed with COSMO-CLM.

  Exactly, therefore here we are using "fully" forced. As in the introduction mentioned above, there is some study specific focusing on one of the external forcings, such as orbital or GHG.

- p4, l135-136: why this association should not be possible simply using a GCM? please better clarify

  The association between the regional climate with large scle circulations is possible but lack of detailed spatial information, therefore the sentence has been modified to: "Further, it enables the exploration of the association between the regional climate patterns at higher horizontal resolution and the large-scale atmospheric circulation patterns from the GCM world."

- p4, l136-138: This is not shown here. I would rather frame it as the possibility to use the results for the study of extreme events on societies. Still, in this case you must make clear that you need a proper comparison against proxy data before using the model results for past times.

  The comparison of the simulated results against proxy record is under processing and planned in another frame of work focus. In this paper we mainly want to compare the climate of the ERP to PI and characterize the ERP climate. So one have pre knowledge about the climate change of the two period and further discussing the trend or variability of the region by linking or comparing the simulation output with proxy records.

- p5, l170: for which area they apply COSMO-CLM for, in the study of Bucchignani? why using their configuration? please specify

  The representation of albedo and aerosols are found to be the most important parameters in the studied region (see also Hartmann et al., submitted) and are set the same as in Bucchignani et al., (2016) in which the simulations are performed over MENA (Middle East and North Africa). In the manuscript we have modifies this part as "MENA region covers similar study region in this study which across the equatorial region, in our previous test simulations, we have found out that the parametrization of albedo and aerosols are the most important ones as described in Bucchignani et al., 2016. Therefore, we have chosen the general model setup from Bucchignani et al., 2016."

- section 2.1: As you mention later in the text, the configuration of the model is very important for an RCM cause it is region-dependent. What was the starting setup of your model? did you use the default setup for Europe provided by the CLM-community? you did not apply any additional changes beside the ones in accordance to Bucchignani et al. 2016? Eventually, provide more context on the reason for your choices in the model setup.

  As mentioned in the previous response, before setting up our transient CCLM simulations, we have performed several test simulations to find the best configuration of the transient simulation. As our simulation domain covers similar region as MENA described in Bucchignani et al., 2016, we have

therebefore ran several test simulations based on the simulation's setup of Bucchignani et al., 2016. The test simulations include several different setups, details in the table shown here:

| Simulations | Specific setting |
| --- | --- |
| CCLM | itype_aerosol=1, soil layer=9 (0.005 - 11.5 m), pstbga= 0.045  ireals / 19330.0  ireals |
| CCLM-MENA | itype_aerosol=2, soil layer=7 (0.005 - 14.58 m), pstbga= 0.045_ireals / 19330.0_ireals |